# Heterochromatin assembly by interrupted Sir3 bridges across neighboring nucleosomes

**Reza Behrouzi[†], Chenning Lu[†], Mark A Currie, Gloria Jih, Nahid Iglesias, Danesh Moazed\***

Department of Cell Biology, Howard Hughes Medical Institute, Harvard Medical School, Boston, United States

**Abstract** Heterochromatin is a conserved feature of eukaryotic chromosomes with central roles in regulation of gene expression and maintenance of genome stability. Heterochromatin formation involves spreading of chromatin-modifying factors away from initiation points over large DNA domains by poorly understood mechanisms. In *Saccharomyces cerevisiae*, heterochromatin formation requires the SIR complex, which contains subunits with histone-modifying, histone-binding, and self-association activities. Here, we analyze binding of the Sir proteins to reconstituted mono-, di-, tri-, and tetra-nucleosomal chromatin templates and show that key Sir-Sir interactions bridge only sites on different nucleosomes but not sites on the same nucleosome, and are therefore 'interrupted' with respect to sites on the same nucleosome. We observe maximal binding affinity and cooperativity to unmodified di-nucleosomes and propose that nucleosome pairs bearing unmodified histone H4-lysine16 and H3-lysine79 form the fundamental units of Sir chromatin binding and that cooperative binding requiring two appropriately modified nucleosomes mediates selective Sir recruitment and spreading.

\*For correspondence: danesh@hms.harvard.edu

[†]These authors contributed equally to this work

**Competing interests:** The authors declare that no competing interests exist.

## Introduction

The packaging of eukaryotic nuclear DNA with histones and other proteins into chromatin is critical for the regulation of transcription, recombination, replication and DNA damage repair. The basic unit of chromatin folding is the nucleosome, in which 147 base pairs of DNA are wrapped around an octamer of two copies each of histones H2A, H2B, H3, and H4 (*Kornberg, 1977*; *Luger et al., 1997*). Site-specific DNA-binding proteins or RNA-based mechanisms recruit histone-modifying proteins that mediate histone posttranslational modifications such as acetylation and methylation (*Holoch and Moazed, 2015*; *Jenuwein and Allis, 2001*; *Kouzarides, 2007*; *Li et al., 2007a*; *Schreiber and Bernstein, 2002*; *Strahl and Allis, 2000*). These modifications provide binding sites for numerous effector proteins that activate or silence transcription and are often associated with large domains of DNA, which can range in size from 2–3 to several hundred kilobases. The assembly of large domains of DNA with activating or repressive histone modifications allows regional and coordinated regulation of gene expression and maintenance of landmark chromosome structures, but the mechanisms that mediate the spreading of histone modification over large DNA domains are poorly understood.

Heterochromatic DNA domains are a conserved feature of eukaryotic chromosomes and provide the most striking examples of regional control (*Moazed, 2001*; *Richards and Elgin, 2002*). Heterochromatin forms at repetitive DNA regions in order to prevent recombination and maintain genome integrity as well as at developmentally regulated genes (*Richards and Elgin, 2002*). Heterochromatin tends to spread from defined initiation sites, leading to the inactivation of genes in a sequence-

**eLife digest** Inside plant, fungi and animal cells, DNA wraps around disc-shaped histone proteins to form structures called nucleosomes. Chains of nucleosomes, each with a small stretch of DNA, help to package meters of genetic material into a compact form called chromatin in the cell's nucleus. Changes to how chromatin is organized can affect how genes switch on and off. Critically, this allows cells to respond to changes in their environment and to develop into the many cell types required to build animals ranging from worms to humans. For example, specialized groups of proteins that bind to nucleosomes, spread along specific sites of chromatin and can change its structure into an inaccessible form called heterochromatin thereby switching off genes. Proteins that bind to specific nucleosomes control the spreading, gene activity, and even memory properties of heterochromatin. However, it is not clear how these proteins spread from their original binding point on the chromatin to other nucleosomes.

Now, Behrouzi, Lu et al. show how heterochromatin spreads to form large, stable structures in budding yeast. Their experiments reveal that heterochromatin proteins attach to sites on neighbouring nucleosomes, forming bridges between them. These findings conflict a long-held view as they show that pairs of nucleosomes, rather than individual nucleosomes, are the natural binding partners for heterochromatin proteins. Also, because these proteins cannot bridge from one side of a nucleosome to the other, they are unlikely to form a continuous chain across multiple nucleosomes on the chromatin. Instead, Behrouzi, Lu et al. observed that a series of short bridges between nucleosomes helps heterochromatin to spread.

To fully understand why bridging only happens between separate nucleosomes, the atomic structure of heterochromatin proteins bound to pairs of nucleosomes needs to be determined. In addition, it will be essential to develop more experimental methods to study the spreading of heterochromatin inside cells.

independent manner (*Talbert and Henikoff, 2006*; *Wang et al., 2014*). The mechanism of spreading of heterochromatin involves the recruitment of chromatin-modifying complexes, which have coupled histone-binding and histone-modifying activities, to specific nucleation sites (*Canzio et al., 2011*; *Grunstein, 1997*; *Hoppe et al., 2002*; *Luo et al., 2002*; *Moazed, 2001*; *Rusche et al., 2002, 2003*). This is then followed by iterative cycles of histone modification and histone binding, which are thought to be coupled with the formation of homo and heterotypic interactions between silencing factors. In these models, silencing factors are proposed to form bridges that span binding sites on the same nucleosome and 'sticky ends' that extend away from nucleation points and mediate interactions across neighboring nucleosomes (*Canzio et al., 2011*; *Moazed, 2001*). The continuous nature of interactions between silencing proteins, both across single nucleosomes and across neighboring ones, amounts to the formation of proteinaceous chromatin-bound oligomers. However, this model has not been tested with chromatin templates that allow the extent to which self-association of silencing factors contributes to specific binding and spreading to be determined. The nature of silencing factor self-associations, how they bridge nucleosomes, and their relationship to the mechanism of spreading therefore remain ambiguous.

Silent chromatin in the budding yeast *S. cerevisiae* serves as a major model system for studies of heterochromatin establishment and inheritance. Silencing at the silent mating type loci (*HML* and *HMR*) and sub-telomeric regions requires silent information regulator (Sir) proteins, Sir2, Sir3, and Sir4, which together form the SIR complex (*Aparicio et al., 1991*; *Klar et al., 1979*; *Liou et al., 2005*; *Moazed et al., 1997*; *Rine and Herskowitz, 1987*; *Rudner et al., 2005*; *Rusche et al., 2003*). During the initiation step, the Sir2 and Sir4 proteins, which together form a stable Sir2/4 heterodimer (*Moazed et al., 1997*), and Sir3, are recruited to the silencer through interactions with silencer-specificity factors, ORC, Abf1, and Rap1 (*Hoppe et al., 2002*; *Luo et al., 2002*; *Moretti et al., 1994*; *Moretti and Shore, 2001*; *Rusche et al., 2002*; *Triolo and Sternglanz, 1996*). The Sir2 subunit, which is an NAD-dependent deacetylase (*Imai et al., 2000*; *Landry et al., 2000*; *Smith et al., 2000*), then deacetylates silencer-proximal nucleosomes, particularly the H4K16 residue, creating a binding site for Sir3 (*Armache et al., 2011*; *Carmen et al., 2002*; *Liou et al., 2005*;

*Wang et al., 2013*). Subsequent iterative cycles of deacetylation and Sir-Sir interactions lead to spreading of SIR complexes (*Hoppe et al., 2002*; *Luo et al., 2002*; *Rusche et al., 2002*) along multiple kilobases of chromatin away from the silencer.

Many of the key activities of the SIR complex have been mapped to specific domains in its subunits and provide important guides for further studies. Modification-sensitive nucleosome binding occurs via a conserved domain at the N terminus of Sir3, called the bromo-adjacent homology (BAH) domain (*Figure 1A*) (*Buchberger et al., 2008*; *Onishi et al., 2007*). The AAA ATPase-like (AAAL) domain of Sir3 also interacts with histones and nucleosomes (*Hecht et al., 1995*). However, this interaction is at least an order of magnitude weaker than the BAH-mediated chromatin interactions (*Martino et al., 2009*; *Wang et al., 2013*). Sir4 forms stable dimers via a coiled-coil domain at its C terminus (Sir4CC), which also forms a binding surface for two Sir3 molecules, linking the Sir2 histone deacetylase to the nucleosome binding subunit of the complex (*Chang et al., 2003*; *Moazed et al., 1997*; *Rudner et al., 2005*). Finally, Sir3 forms dimers via a winged helix (wH) domain at its C terminus (*Oppikofer et al., 2013*). Although all of the above interaction domains are critical for silencing, how they promote the spreading of silencing remains unknown.

In this study, we use equilibrium and kinetic binding experiments to compare the association of Sir3 and its subfragments with in vitro reconstituted mono-, di-, tri-, and tetra-nucleosomal chromatin templates (MonoN, DiN, TriN, and TetraN, respectively), and determine how this association is affected by the Sir4 coiled-coil (Sir4CC) domain. Our analysis shows that, at physiological concentrations, Sir3 binds to DiN with maximal cooperativity by a mechanism that requires the Sir3wH dimerization domain and is enhanced by the Sir4CC domain. In contrast, although each nucleosome contains two Sir3 binding sites, the association of Sir3 with MonoN, in the presence or absence of Sir4CC, occurs by a non-cooperative mechanism, suggesting that these interactions mediate lateral Sir3-Sir3 bridging across two nucleosomes, rather than on the same nucleosome. Moreover, we show that H4K16 acetylation and H3K79 trimethylation, two well-established anti-silencing modifications, work together to dramatically reduce the affinity of Sir3 for nucleosomes. Together, our findings suggest that spreading of the Sir proteins on chromatin involves the cooperative recruitment of new SIR complexes to pairs of nucleosomes lacking H4K16 acetylation and H3K79 methylation independently of interactions with already bound SIR complexes. This inter-nucleosomal cooperative mode of binding suggests that interrupted Sir3 bridges across neighboring nucleosomes, stabilized by Sir4, are the primary driving force for heterochromatin spreading.

## Results

### Sir3 binds to DiN with high affinity and cooperativity

Sir3 molecules self-associate with high affinity ($K_D$ ~2 nM, *Liou et al., 2005*) to form mostly homo-dimers and to a lesser extent oligomers (*Figure 1A*) (*Liou et al., 2005*; *McBryant et al., 2006*). Due to the multivalent nature of Sir3 dimers, potential interactions between Sir3 proteins bound to the same (intra-nucleosomal bridging) or adjacent nucleosomes (inter-nucleosomal bridging) may cooperatively stabilize Sir3 interactions with properly modified chromatin (*Figure 1B*). These different modes of binding predict different Sir3 affinities for mono- and di-nucleosomes (MonoN and DiN, respectively) that will depend on Sir3 dimerization. In order to investigate the mechanism of Sir3 binding to chromatin and to distinguish intra- and inter-nucleosomal contributions to Sir3-chromatin association, we studied binding of Sir3 with MonoN and DiN.

We reconstituted defined nucleosome arrays by the salt-gradient dialysis method, as described previously (*Huynh et al., 2005*; *Luger et al., 1999*), using the 601 nucleosome positioning sequence and *S. cerevisiae* histones purified from *E. coli* (*Figure 1—figure supplement 1A–E*). The quality of reconstituted nucleosomes was assessed by electrophoretic mobility shift assays (EMSA) and restriction enzyme protection (*Figure 1—figure supplement 1F*). Purified overexpressed Sir3 from *S. cerevisiae* (*Liou et al., 2005*) was used for all assays (*Figure 1—figure supplement 1G*) to maintain the Nα-acetylation of Sir3, which is required for its efficient binding to nucleosomes (*Arnaudo et al., 2013*; *Onishi et al., 2007*; *Yang et al., 2013*) and silencing in vivo (*Wang et al., 2004*). Various purification strategies resulted in Sir3 proteins with identical purity and nucleosome binding behavior (see Materials and methods). Binding experiments were performed in salt concentrations that had

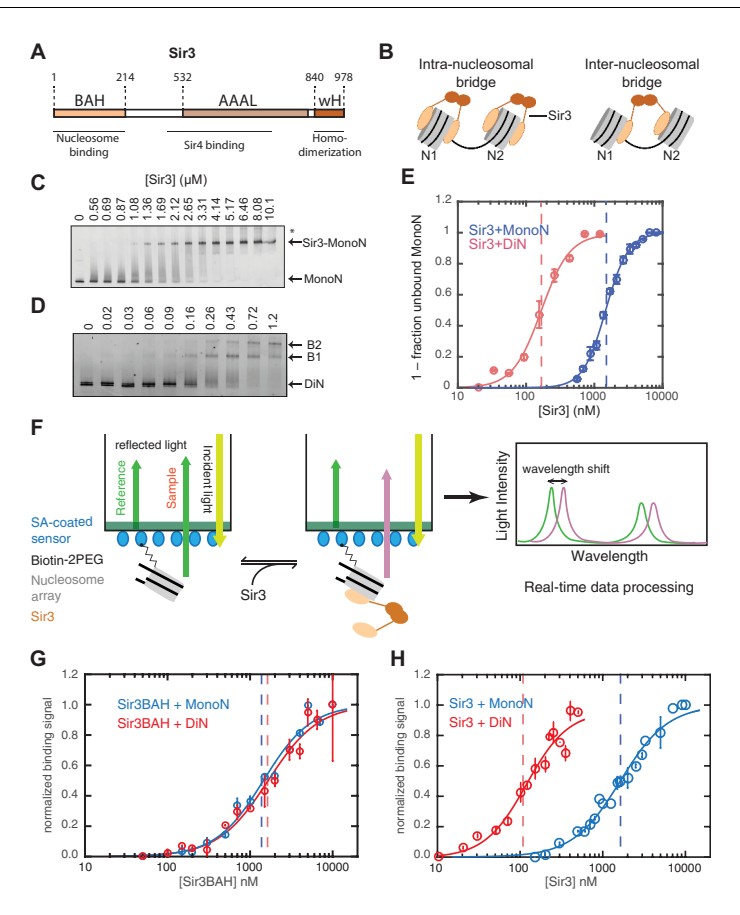

**Figure 1.** Cooperative association of Sir3 with DiN. (**A**) Schematic diagram of the Sir3 primary sequence showing the location of the BAH, AAAL, and wH domains. (**B**) Models for the association of Sir3 dimers with chromatin. (**C, D**) Representative EMSA showing Sir3 binding to unmodified MonoN (**C**) and DiN (**D**). Purified Sir3 proteins were titrated onto a constant amount of MonoN or DiN at 3 nM. Samples were separated on native gels, nucleosomes were stained with SYBR Gold, and the amount of unbound nucleosomes was quantified by the staining intensity of the unshifted nucleosome band. *, higher mobility shifted band that may result from either bridging of MonoN by Sir3 or other minor high molecular weight Sir3-MonoN complexes. Band 1 (B1) likely reflects Sir3-DiN in bridged conformation, whereas Band 2 (B2) shows additional binding to single nucleosome surfaces. (**E**) Quantification and analysis of Sir3 binding to MonoN. Binding curves from three experiments performed as in (**C**) and (**D**) were fitted with the Hill Equation. The apparent $K_D$ values for Sir3 binding to MonoN (blue dotted line) and DiN (red dotted line) are indicated. (**F**) Bio-layer interferometry (BLI) assay detects changes in binding of proteins to surface-immobilized nucleosome templates by measuring the wavelength shift of the light reflected from the surface. (**G, H**) Binding of Sir3BAH (a.a. 1–382) (**G**) and full-length Sir3 (**H**) to MonoN and DiN after background correction and normalization to min-max of binding signal was fit with the Hill equation (see Materials and methods and **Table 1**). Data from 2–3 replicate experiments (>30 data points) (**G**) and from 3 or more replicate experiments (>30 data points) (**H**) were pooled for model fitting. The apparent $K_D$ for Sir3BAH and Sir3 binding to MonoN (blue) and DiN (red) is indicated. See **Table 1** for parameter values.

The following figure supplements are available for figure 1:

**Figure supplement 1.** In vitro reconstitution of MonoN, DiN, TriN, and TetraN.

**Figure supplement 2.** Association of the Sir3 with DiN and Sir3BAH domain with MonoN and DiN.

**Figure supplement 3.** Measurements of Sir3 binding to MonoN with BLI.

**Figure supplement 4.** Analysis of BLI binding profiles.

previously been determined to render Sir3 binding strongly sensitive to H4K16Q mutation, resembling the in vivo behavior of Sir3 (*Johnson et al., 2009*; *Swygert et al., 2014*).

To determine the affinity of Sir3 for MonoN and DiN, we performed EMSA (*Buchberger et al., 2008*; *Johnson et al., 2009*; *Liou et al., 2005*). We found that Sir3 bound to MonoN with a $K_D$ around 1.7 µM (*Figure 1C,E*) and to the DiN with a $K_D$ around 0.17 µM (*Figure 1D,E*), indicating a ~10 fold increase in affinity for DiN relative to MonoN, much higher than what would be expected by the increase in the number of independent binding sites. ScaI digestion of the linker DNA in the DiN resulted in a reduction in binding affinity to that observed for the MonoN (*Figure 1—figure supplement 2A*), indicating that specific binding to DiN, not extra DNA content, is responsible for higher affinity binding (Other assays ruling out DNA binding are described in Materials and methods). In contrast, Sir3BAH domain bound to DiN with only around 2-fold higher affinity than to MonoN (*Figure 1—figure supplement 2B,C*; *Table 1*), as expected from the higher number of binding sites on the DiN.

Although EMSA allows for direct observation of complex formation and may even elucidate assembly intermediates, it is a quasi-equilibrium method, which might be affected by fast dynamics of the complex in the gel matrix or spurious Sir3-nucleosome interactions due to the low ionic strength and temperature of EMSA buffer. To validate the EMSA observations using an equilibrium assay, we examined Sir3-nucleosome interactions at physiological ionic strength and temperature using the BioLayer Interferometry (BLI) assay (*Abdiche et al., 2008*). To perform BLI measurements, we immobilized biotinylated MonoN or DiN (*Figure 1—figure supplement 3A,B*) on the surface of streptavidin-coated biosensors and studied changes in the number of Sir3 molecules bound to nucleosomes by monitoring in real time the wavelength shifts of the reflected light from the biosensor surface (*Figure 1F*, see Materials and methods). The binding signal at equilibrium reflects the number of Sir3 molecules bound to nucleosomes at any given Sir3 concentration. We reconstructed binding

**Table 1.** Thermodynamic parameters describing the binding of Sir3 protein to nucleosomes. Data from more than 2 replicate titration experiments were pooled and the BLI data were fit with Hill equation (see Materials and methods). Uncertainties show 68% confidence intervals around fit parameters (±1 SD) reported by fitting algorithm.

| Binding experiments | | BLI | | EMSA[*] |
| --- | --- | --- | --- | --- |
| | | *Apparent $K_D$ (µM)* | *Hill coefficient* | *Apparent $K_D$ (µM)* |
| MonoN | Sir3 | 1.4 ± 0.06 | 1.3 ± 0.1 | 1.7 ± 0.20 |
| | Sir3+Sir4CC | N/A | N/A | 1.4 ± 0.10 |
| | Sir3ΔwH | 1.2 ± 0.10 | 0.93 ± 0.07 | 1.0 ± 0.10 |
| | Sir3ΔwH+Sir4CC | N/A | N/A | 0.9 ± 0.05 |
| | Sir3BAH | 1.4 ± 0.10 | 1.5 ± 0.2 | 2.1 ± 0.20 |
| DiN | Sir3 | 0.12 ± 0.01 | 1.9 ± 0.2 | 0.17 ± 0.10 |
| | Sir3+Sir4CC | N/A | N/A | 0.08 ± 0.01 |
| | Sir3ΔwH | 1.1 ± 0.05 | 1.2 ± 0.1 | 0.62 ± 0.10 |
| | Sir3ΔwH+Sir4CC | N/A | N/A | 0.12 ± 0.01 |
| | Sir3BAH | 1.6 ± 0.10 | 1.4 ± 0.1 | 1.40 ± 0.20 |
| Sir3 | acMonoN | N/A | N/A | 4.0 ± 0.20 |
| | meMonoN | N/A | N/A | 5.2 ± 0.20 |
| | acDiN | N/A | N/A | 0.7 ± 0.05 |
| | meDiN | N/A | N/A | 0.8 ± 0.05 |
| | ac/meMonoN | N/A | N/A | >11[†] |
| | ac/meDiN | N/A | N/A | >3[†] |

[*]Hill coefficients obtained from EMSA appeared unreliable due to assay artifacts, such as non-specific binding to DNA, and are not reported.

[†] Nonspecific binding could not be measured accurately due to low affinity.

isotherms by plotting normalized binding signals at equilibrium *vs.* Sir3 concentration (see Materials and methods for further details). As controls, binding of Sir3 to immobilized nucleosomes was insensitive to whether the biotin moiety was attached to histone H2A via a flexible 30 Å linker or to the end of nucleosomal DNA with a 20 bp extension (*Figure 1—figure supplement 3C*). In contrast, it was strongly sensitive to the acetylation of nucleosomes (*Figure 1—figure supplement 3D*), indicating that the BLI signal reflected specific Sir3-nucleosome interactions.

We observed that Sir3BAH bound to MonoN and DiN with the nearly identical apparent affinity of ~1.4 µM (*Figure 1G*, *Table 1*), consistent with EMSA measurements (*Table 1*). Similarly, analysis of the binding profiles of full-length Sir3 to MonoN and DiN resulted in calculated apparent $K_D$ values of around 1.4 and 0.11 µM, respectively (*Figure 1H*, *Table 1*), which are similar to those obtained by EMSA (*Table 1*). The Hill coefficient of Sir3 and Sir3BAH binding to MonoN was nearly identical and close to 1 (*Figure 1G*, *Table 1*). In fact, binding of Sir3 or Sir3BAH to MonoN could be satisfactorily described by a simple model based on saturation of two identical and independent (i.e. non-cooperative) binding sites with a macroscopic dissociation constant of 1.2–1.4 µM (see Materials and methods), confirming the absence of cooperativity in binding of Sir3 to MonoN. In contrast, we observed cooperative binding of Sir3 to DiN, reflected in the Hill coefficient of ~2 (*Figure 1H*; *Table 1*). Therefore, the complex of Sir3 with DiN, but not with MonoN, is stabilized by a cooperative mechanism, supporting the inter-nucleosomal mode of bridging depicted in *Figure 1B*.

## DiN units constitute the fundamental unit of chromatin for Sir3 binding

Our findings above show that Sir3-Sir3 contacts across different nucleosomes (*Figure 1B*, right) play an important role in stabilizing Sir3-chromatin complexes. In contrast, even though there are two binding surfaces on each nucleosome for Sir3, intra-nucleosomal Sir3 interactions are prohibited (*Figure 1B*, left). Therefore, one may conclude that in the context of chromatin arrays, unmodified DiN units act as independent high affinity binding sites for Sir3 dimers. To directly test this hypothesis, we compared the binding of Sir3 to DiN versus larger nucleosomal arrays using the BLI assay. Positive interactions between binding sites, beyond those present in the DiN template (equivalent to Sir3 oligomerization on chromatin), would result in higher apparent binding affinity and cooperativity for longer nucleosome arrays compared to DiN. In stark contrast to the difference in binding affinity between MonoN and DiN templates (*Figure 1E,H*), Sir3 binding to tri- and tetra-nucleosome templates (TriN and TetraN, respectively) displayed very similar binding affinity and cooperativity as binding to DiN (*Figure 2A*). This result suggests that even in the context of nucleosome arrays Sir3-chromatin interactions that contribute to binding stability are limited to sites on only two different nucleosomes (*Figure 1B*, right). We therefore conclude that binding sites on two different nucleosomes form the fundamental unit of chromatin binding for Sir3 dimers.

## Sir3 cooperative binding to DiN is mediated by its C-terminal wH dimerization domain

Sir3 forms dimers and oligomers in vitro (*King et al., 2006*; *Liou et al., 2005*; *McBryant et al., 2006*; *Moretti et al., 1994*), and its C-terminal winged helix (wH) domain is necessary and sufficient for dimerization (*Oppikofer et al., 2013*) (see *Figure 1A* for Sir3 domains). Furthermore, deletion of the wH domain abolishes Sir3 association with silent chromatin regions and disrupts silencing at both the mating-type loci and telomeres (*Oppikofer et al., 2013*). We therefore investigated whether and how the wH domain may contribute to the cooperative mechanism of Sir3 binding to DiN by studying the binding of affinity purified Sir3 lacking the wH domain (Sir3ΔwH) (*Figure 2—figure supplement 1A*) to both MonoN and DiN.

Both EMSA and BLI assays showed that Sir3ΔwH bound to MonoN with a $K_D$ around 1.1 µM (*Figure 2A,B* and *Figure 2—figure supplement 1B*, *Table 1*), similar to the $K_D$ value (1.4 µM) we observed for the association of full-length Sir3 with MonoN. Deletion of the wH domain therefore did not affect Sir3 affinity for MonoN. Moreover, in contrast with ~10 fold increase in the apparent affinity of full-length Sir3 for DiN compared to MonoN (*Figure 1G,H*, *Table 1*), we did not observe a significant difference between binding of Sir3ΔwH to DiN and MonoN (*Figure 2B,C*, *Table 1*). Further analysis of the BLI binding data with the Hill equation revealed that in contrast to full-length Sir3 (*Figure 1G,H*), Sir3ΔwH, like Sir3BAH, bound to MonoN and DiN without cooperativity (*Table 1*).

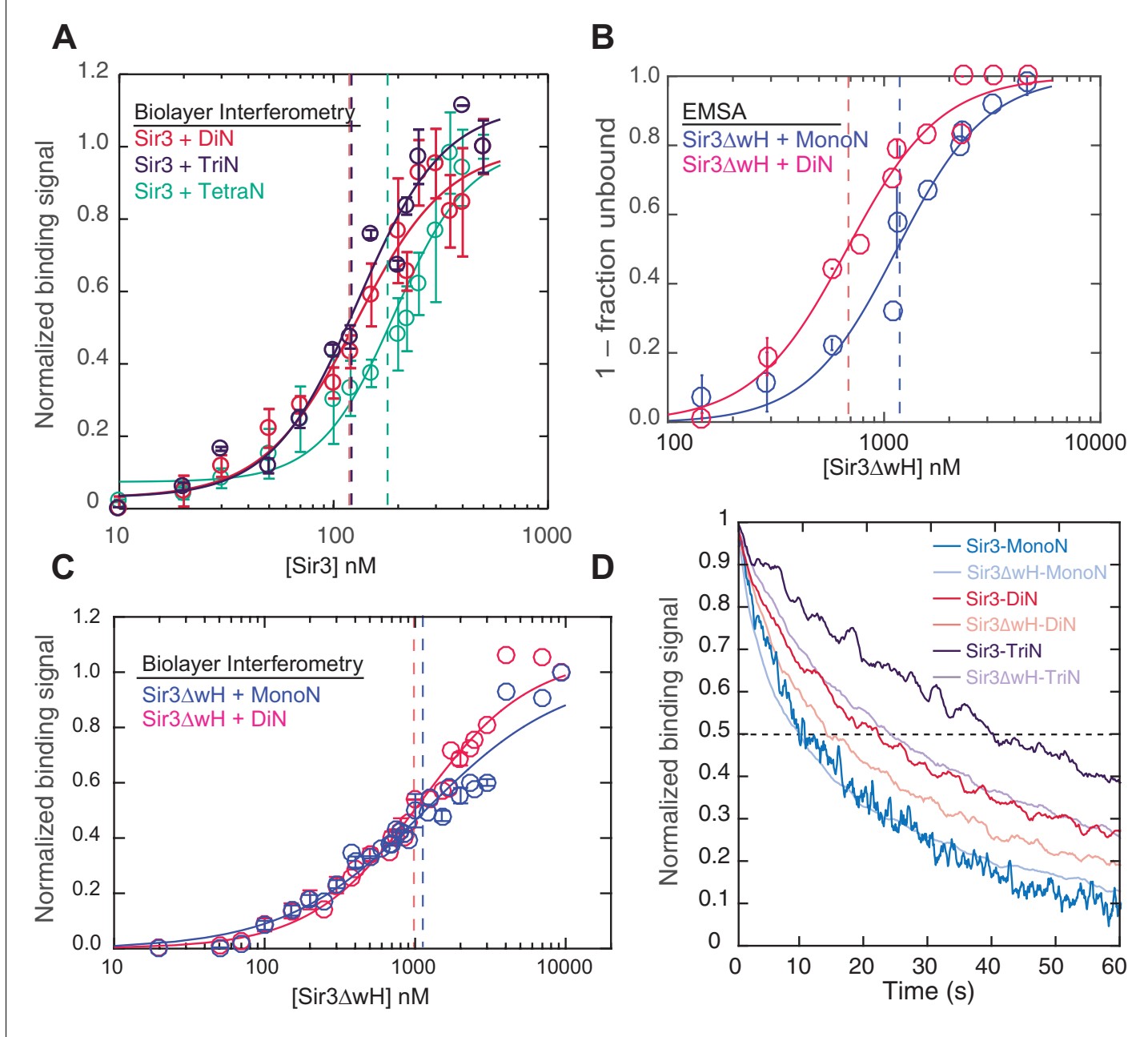

**Figure 2.** Sir3 displays maximal cooperative binding to DiN that is mediated by its winged helix (wH). (A) Binding of full-length Sir3 to DiN, TriN, and TetraN templates after background correction and normalization to min-max of binding signal was fit with the Hill equation (see Materials and methods and *Table 1*). Data from 3 or more replicate experiments (>30 data points) were pooled for model fitting. Vertical dotted line indicates the apparent $K_D$ for Sir3 binding to DiN, TriN and TetraN. (B) EMSA showing the binding of Sir3ΔwH to MonoN (blue) and DiN (red). Binding curves from three experiments performed (see *Figure 2—figure supplement 1*) were fitted with the Hill Equation. (C) Sir3ΔwH MonoN and DiN binding data from BLI assays normalized to the range of binding signal and fit with Hill equation (see Materials and methods). Data from two independent replicates were pooled before model fitting. (D) Kinetics of Sir3-nucleosome complex dissociation measured by the BLI assay reveals that the Sir3wH domain is required for the cooperative stabilization of Sir3-DiN and Sir3-triN complexes. Representative dissociation profiles obtained at 0.2 μM Sir3 were self-normalized for easier visual comparison. See *Table 1* for parameter values.

The following figure supplement is available for figure 2:

**Figure supplement 1.** Purification of Sir3ΔwH and EMSA assays with Sir3ΔwH.

We therefore conclude that the wH dimerization domain is required for Sir3 cooperative and high affinity binding to DiN, without affecting Sir3 binding to MonoN. Furthermore, measurement of complex dissociation rates by BLI revealed that loss of wH domain reduced the half-life of Sir3-DiN and Sir3-TriN complexes by ~40%, while having little or no effect on Sir3-MonoN complex half-life (*Figure 2D*, *Table 2*). These findings support our conclusion regarding Sir3 dimerization in forming inter-nucleosomal bridges and the absence of intra-nucleosomal Sir3 contacts, and suggest that inter-nucleosomal bridges may contribute to forming temporally stable heterochromatin domains in vivo.

Furthermore, Sir3 proteins lacking both the AAAL and wH domains (Sir3BAH), but not wH domain alone (Sir3ΔwH), displayed drastically reduced Sir3-MonoN complex half-life (*Figure 1—figure supplement 4A*, see also Materials and methods). Therefore, although AAAL domain on its own interacts weakly with chromatin (*Wang et al., 2013*), it plays an important role in stabilizing the BAH-

**Table 2.** Kinetic parameters describing the binding of Sir3 protein to nucleosomes. Rates and amplitudes, representing the average of 3 or more measurements at different Sir3 concentrations in the ranges specified below, were obtained by fitting data with appropriate rate equations (Suppl. Materials and methods). Values in parentheses indicate standard deviations. In the low range of concentrations, binding rates to mono- and di-nucleosomes were within the expected range of diffusion-limited rate of protein interactions (*Schreiber et al., 2009*). At higher concentrations, however, binding proceeded with rates slower than diffusion limit, probably due to the competition with other modes of binding to nucleosome surfaces.

| Association | | | $A_{obs, 1}$ | | $k_{obs, 1}$ (s$^{-1}$) | | $A_{obs, 2}$ | | $k_{obs, 2}$ (s$^{-1}$) | |
|---|---|---|---|---|---|---|---|---|---|---|
| [Sir3] < 0.3 μM | MonoN | Sir3 | 51% | (11%) | 0.25 | (0.07) | 49% | (11%) | 0.04 | (0.01) |
| | | Sir3ΔwH | 73% | (11%) | 0.14 | (0.06) | 27% | (11%) | 0.03 | (0.01) |
| | | Sir3BAH | 100% | (0%) | 0.65 | (0.34) | 0% | (0%) | N/A | N/A |
| | DiN | Sir3 | 100% | (0%) | 0.06 | (0.01) | 0% | (0%) | N/A | N/A |
| | | Sir3ΔwH | 11% | (22%) | 0.16 | N/A | 89% | (22%) | 0.08 | (0.01) |
| | | Sir3BAH | 100% | (0%) | 0.40 | (0.18) | 0% | (0%) | N/A | N/A |
| [Sir3] 1.5–4 μM | MonoN | Sir3 | 56% | (1%) | 0.62 | (0.02) | 44% | (1%) | 0.08 | (0.01) |
| | | Sir3ΔwH | 74% | (4%) | 0.70 | (0.07) | 26% | (4%) | 0.10 | (0.02) |
| | | Sir3BAH* | 100% | (0%) | 0.53 | (0.06) | 0% | (0%) | N/A | N/A |
| | DiN | Sir3 | 55% | (2%) | 0.51 | (0.10) | 45% | (2%) | 0.08 | (0.00) |
| | | Sir3ΔwH | 56% | (3%) | 0.47 | (0.05) | 44% | (3%) | 0.09 | (0.02) |
| | | Sir3BAH* | 100% | (0%) | 0.46 | (0.03) | 0% | (0%) | N/A | N/A |

| | | | $i$ [†] | $A_{off, 1}$ | $\tau_{off, 1}$(S) | $A_{off, 2}$ | $\tau_{off, 2}$ (S) | $k_{on,1}$ (M$^{-1}$s$^{-1}$)[‡] | | $k_{on,2}$ (M$^{-1}$s$^{-1}$)[‡] | |
|---|---|---|---|---|---|---|---|---|---|---|---|
| [Sir3] < 0.3 μM | MonoN | Sir3 | 2 | 38% (3%) | 7.4 (3.5) | 62% (3%) | 78.5 (23.7) | 2.4E + 05 | (6.5E + 4) | 6.9E + 04 | (2.0E + 4) |
| | | Sir3ΔwH | 2 | 56% (2%) | 8.4 (0.9) | 44% (2%) | 75.3 (3.6) | 1.1E + 0 | (4.3E + 4) | 3.6E + 04 | (1.4E + 4) |
| | | Sir3BAH | 2 | 100% (0%) | 5.6 (1.9) | 0% (0%) | N/A N/A | 1.4E + 06 | (9.2E + 5) | N/A | N/A |
| | DiN | Sir3 | 1 | 92% (14%) | 51.6 (3.8) | 8% (14%) | N/A N/A | 3.8E + 05 | (6.8E + 4) | N/A | N/A |
| | | Sir3ΔwH | 4 | 52% (2%) | 10.8 (1.2) | 48% (2%) | 83.8 (6.5) | 9.7E + 04 | (3.7E + 4) | 4.4E + 04 | N/A |
| | | Sir3BAH | 4 | 100% (0%) | 5.0 (1.4) | 0% (0%) | N/A N/A | N/A | N/A | N/A | N/A |
| [Sir3] 1.5–4 μM | MonoN | Sir3 | 2 | 52% (3%) | 7.7 (0.5) | 48% (3%) | 82.2 (3.7) | 9.1E + 04 | (2.4E + 4) | 1.3E + 04 | (4.0E + 3) |
| | | Sir3ΔwH | 2 | 69% (2%) | 3.4 (0.5) | 31% (2%) | 44.0 (12.4) | 8.5E + 04 | (3.2E + 4) | 1.6E + 04 | (7.3E + 3) |
| | | Sir3BAH | 2 | 100% (0%) | 5.2 (0.5) | 0% (0%) | N/A N/A | 5.5E + 04 | (1.3E + 4) | N/A | N/A |
| | DiN | Sir3 | 4 | 42% (3%) | 5.8 (0.1) | 58% (3%) | 48.6 (2.8) | 3.7E + 04 | (8.1E + 3) | 7.2E + 03 | (3.5E + 3) |
| | | Sir3ΔwH | 4 | 63% (1%) | 5.2 (0.6) | 37% (1%) | 48.5 (3.8) | 3.2E + 04 | (7.5E + 3) | 8.7E + 03 | (2.9E + 3) |
| | | Sir3BAH | 4 | 100% (0%) | 5.2 (0.5) | 0% (0%) | N/A N/A | 2.1E + 04 | (7.1E + 3) | N/A | N/A |

* Slow dissociation phase was not quantified due to small amplitude.

[†] Presumed number of binding sites used in the calculation of $k_{on}$.

[‡] Rates are calculated from ($k_{obs,1}$, $k_{off,1}$) and ($k_{obs,2}$, $k_{off,2}$) pairs.

mediated Sir3-nucleosome complex. This effect is distinct from the wH domain-mediated stabilization of Sir3 inter-nucleosomal bridges discussed above.

## Sir4CC stabilizes the Sir3 bridge across the DiN

Sir4 forms dimers, via its C-terminal coiled-coil (CC) domain, and this dimerization activity is required for silencing at both telomeres and the mating-type locus (*Figure 3A*) (*Chang et al., 2003*; *Chien et al., 1991*; *Murphy et al., 2003*). Sir4CC also interacts with Sir3, through the Sir3 AAA ATPase-like (AAAL) domain, and this interaction is required for Sir3 recruitment and silencing in vivo (*Figure 3A*) (*Chang et al., 2003*; *Ehrentraut et al., 2011*; *King et al., 2006*; *Rudner et al., 2005*). We therefore speculated that Sir4CC might further stabilize Sir3-nucleosome interactions. There are at least two possible ways that Sir4 may interact with Sir3-nucleosome complexes. In the first mode, the Sir4CC mediates interactions between Sir3 molecules bound to each side of the same nucleosome forming an intra-nucleosomal bridge (*Figure 3B*, left). In the second model, the Sir4CC interacts with Sir3 molecules bound to two neighboring nucleosomes, adding a second layer of inter-nucleosomal interactions (*Figure 3B*, right). Similar to the different Sir3 binding modes described above, these two models predict different Sir4CC effects on the binding affinity of Sir3 for the MonoN and DiN, and therefore can be distinguished by binding experiments. Intra-nucleosomal bridging predicts that Sir3/Sir4CC has a higher binding affinity towards MonoN than Sir3 alone. In contrast, inter-nucleosomal bridging predicts (1) higher binding affinity of Sir3/Sir4CC towards DiN compared with Sir3 alone, because of the dimerizing Sir3wH domain and the interaction of Sir4CC with Sir3 AAAL domains, and (2) no change in the binding affinity of Sir3 towards MonoN upon the addition of Sir4CC.

EMSA experiments showed that although the addition of Sir4CC caused a slight upshift of Sir3-MonoN band, it did not affect Sir3-MonoN binding affinity, as the $K_D$ values were similar with or without Sir4CC (*Figure 3C,E*, *Table 1*). In contrast, Sir4CC decreased the apparent $K_D$ value of Sir3 binding to DiN about 2 fold, from 0.17 to 0.08 μM (*Figure 3D,F*, *Table 1*), suggesting that Sir4 binds to Sir3 proteins that bridge neighboring nucleosomes. Consistent with the above binding results (*Figure 4C–F*), Sir4CC did not affect the binding affinity of Sir3ΔwH towards MonoN (*Figure 3G*, *Figure 3—figure supplement 1A,C*, *Table 1*), but increased its binding affinity towards DiN about 4-fold (*Figure 3H*, *Figure 3—figure supplement 1B,C*, *Table 1*). We conclude that both Sir3wH and Sir4CC dimerization domains contribute to inter-nucleosomal bridging, but they may perform partially redundant functions in this respect.

## Sir3 crosslinks mono-nucleosomes in solution

Since Sir3 binds to DiN cooperatively, we tested whether it could act as a bridge linking free MonoN in solution. To achieve this, we devised a crosslinking assay in which the ability of a nucleosome immobilized on a solid resin to bind to a free nucleosome could be tested (*Figure 4A*). We assembled MonoN with 5' biotinylated 601 DNA containing a 20 bp linker 5' to the 601 sequence to allow sufficient space and flexibility of the nucleosome away from the solid support. The reconstituted biotinylated nucleosome was conjugated to streptavidin magnetic beads and incubated with Sir3 or Sir3ΔwH, either alone or in combination with Sir4CC (*Figure 4A*). This was followed by incubation with Alexa-647-labeled MonoN. The beads and their associated proteins and nucleosomes were then recovered by magnetic concentration and washed prior to elution of nucleosomal DNA from the beads with 2 M NaCl. The resulting supernatant was analyzed by gel electrophoresis, and the amount of pulled down Alexa-647 nucleosomal DNA was quantified by the intensity of its fluorescent band.

The addition of Sir3 to immobilized nucleosomes promoted the recovery of free labeled nucleosomes and this recovery was not affected by the addition of Sir4CC (*Figure 4B*, lanes 4, 5, and 6; *Figure 4D*), even at lower Sir3 concentrations that still mediated substantial bridging (*Figure 4E*). In contrast, Sir3ΔwH, alone or in the presence of Sir4CC did not promote the recovery of free nucleosomes (*Figure 4B*, lanes 7–9; *Figure 4D*). Western Blot analysis showed that similar amounts of proteins were loaded onto the resin (*Figure 4C*), ruling out the possibility that differences in bridging activity of different proteins were caused by unequal loading of proteins onto nucleosomes. Therefore, this result indicated that Sir3, through its wH domain, acts as a bridge linking mono-nucleosomes in solution. We note that the inability of Sir4CC to stimulate bridging, despite its effect in

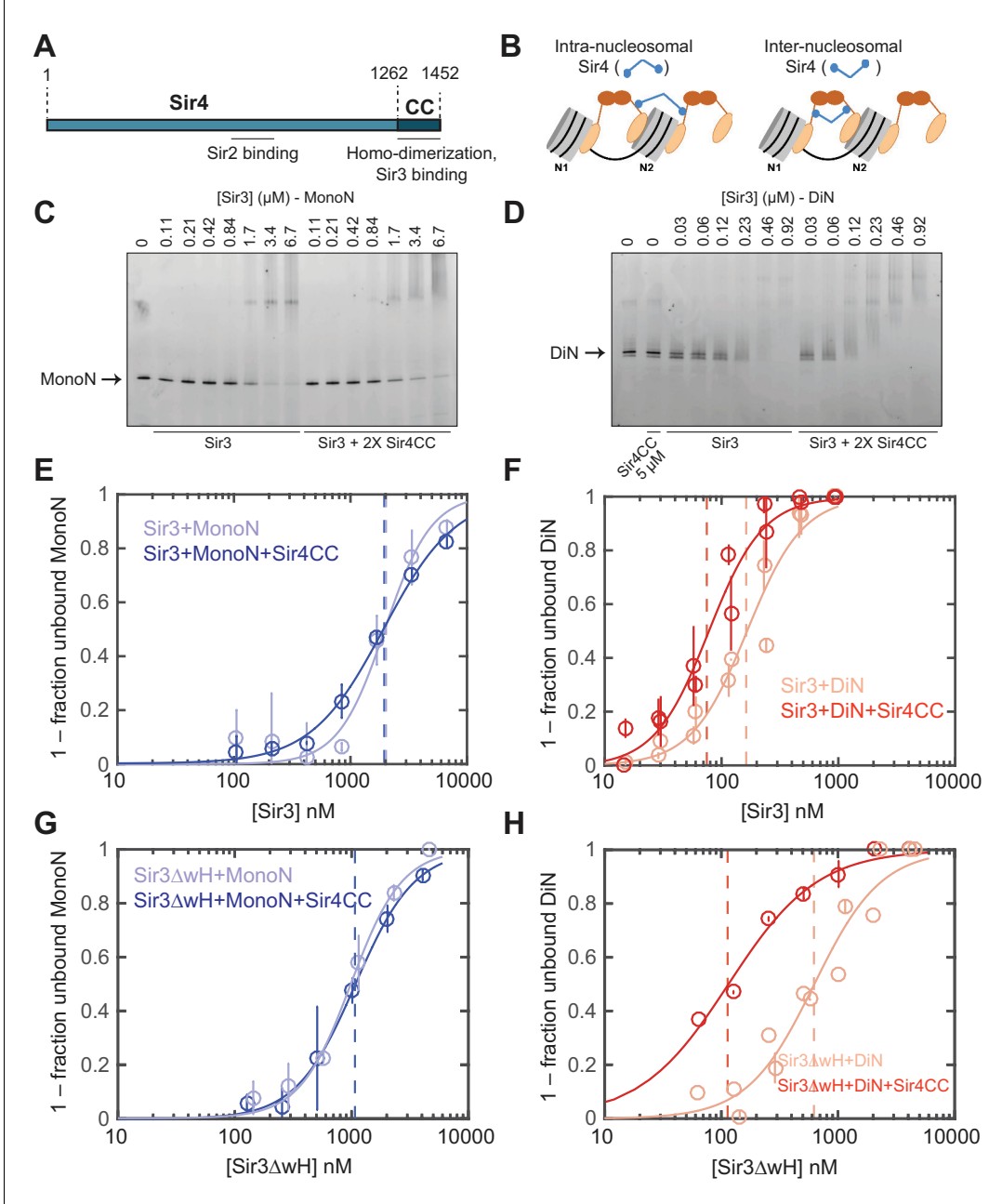

**Figure 3.** Sir4CC does not affect Sir3 binding to MonoN, but increases its affinity towards DiN. (A) Schematic diagram of Sir4 primary sequence showing the location of the coiled-coil (CC) domain and the Sir2 interaction domain (aa747–893). (B) Models for the association of Sir4 with Sir3-bound nucleosomes. (C, D) EMSA experiments comparing Sir3 binding to MonoN (C) and DiN (D) in the presence or absence of Sir4CC. (E, F) Binding curves from three experiments performed as in (C) and (D), respectively, were fitted with the Hill Equation. (G, H) Comparison of Sir3ΔwH binding to MonoN (G) and DiN (H) in the presence or absence of Sir4CC. Binding curves from three experiments performed as in *Figure 3—figure supplement 1A and B* were fitted with the Hill equation.

The following figure supplement is available for figure 3:

**Figure supplement 1.** EMSA assays with Sir3ΔwH and Sir4CC.

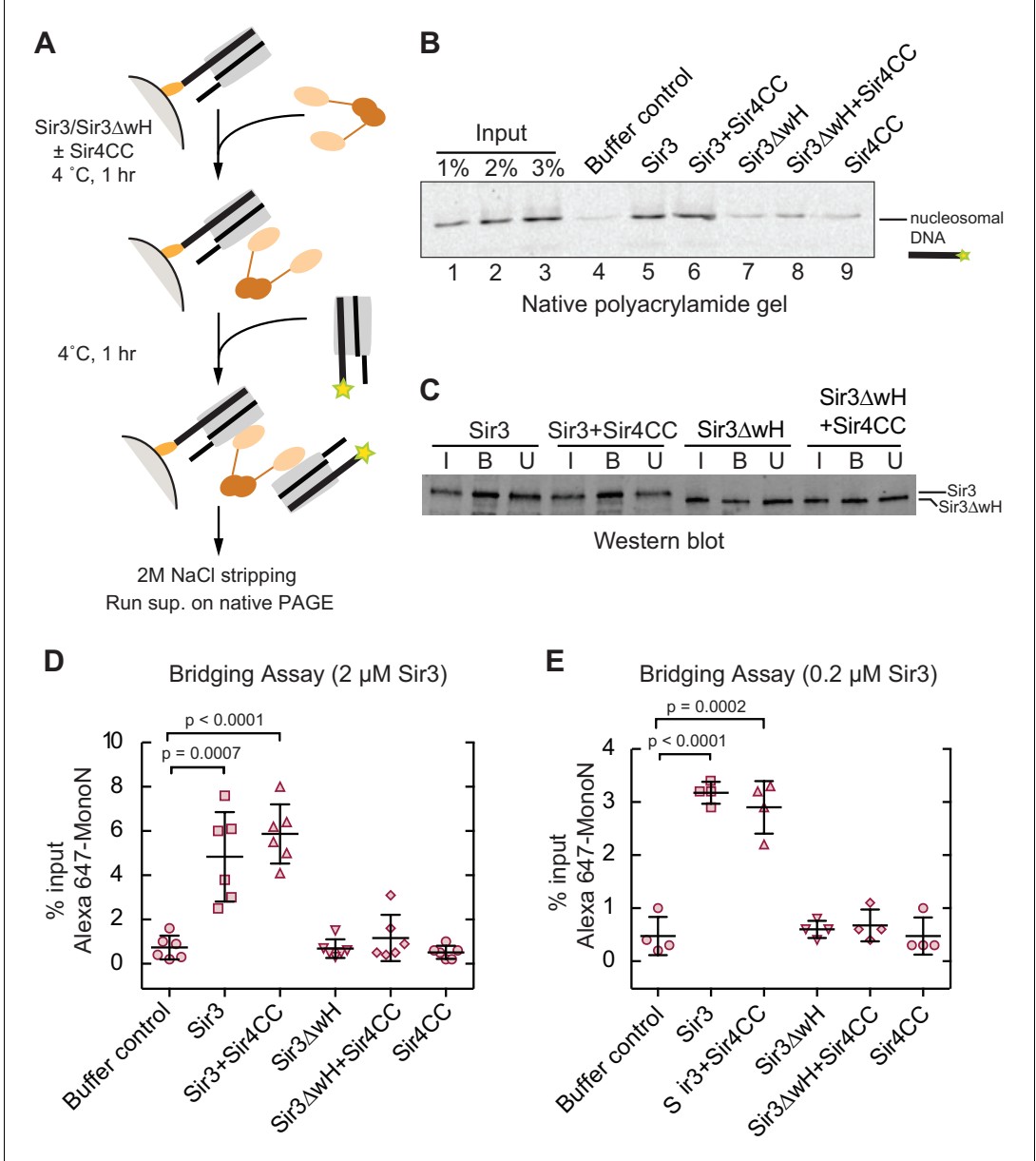

**Figure 4.** Sir3 crosslinks free mono-nucleosomes in solution. (**A**) Illustration of the crosslinking assay. (**B**) A representative native polyacrylamide gel showing the fluorescent DNA pulled down from different reaction mixtures. (**C**) A representative western blot showing Sir3 and Sir3ΔwH in all reactions bound to nucleosomes efficiently. I, input; B, bound, U, unbound. (**D**) Percentage of input Alexa 647- nucleosomes that was pulled down in different reaction mixtures where 2 μM Sir3 or Sir3ΔwH protein concentration was used. Quantification for 6 experiments is presented. (**E**) Same as **D** but using 0.2 μM Sir3 or Sir3ΔwH protein. Quantification for 4 experiments is presented.

increasing the affinity of Sir3 and Sir3ΔwH binding to DiN (*Figure 3F,H*), may suggest that Sir3-Sir4CC interactions are too weak to endure the stringent test of the pull-down assay. In this assay, the concentration of nucleosomes relative to each other is much lower (100 nM) than the effective local nucleosome concentration in the DiN EMSA. Furthermore, weak or highly dynamic complexes may partially disassemble during the wash steps of the assay. We suspect that while Sir4CC can readily link Sir3 bound to adjacent DNA-linked nucleosomes, the contact is not strong enough by itself to stably bring together separate Sir3-bound nucleosomes. A weak Sir4CC-Sir3 interaction in solution is consistent with most Sir3 not being associated with Sir4 in yeast extracts (*Moazed et al.,*

*1997*; *Rudner et al., 2005*). In contrast, the wH-wH interactions can mediate relatively stable dimers in solution (*Oppikofer et al., 2013*).

## H4K16 acetylation and H3K79 trimethylation act together to inhibit Sir3 binding to nucleosomes

Both H4K16 and H3K79 play important roles in silencing in *S. cerevisiae* (*Braunstein et al., 1993*; *Johnson et al., 1990*; *Ng et al., 2002*; *van Leeuwen et al., 2002*). Previous work showed that H4K16 acetylation (H4K16ac) and H3K79 methylation (H3K79me) each inhibits Sir3 binding to histone peptides and nucleosomes, but the difference in binding constants between unmodified and singly modified nucleosomes is rather modest (*Johnson et al., 2009*; *Martino et al., 2009*; *Swygert et al., 2014*; *Wang et al., 2013*). We quantified the effect of H4K16ac and H3K79me and, more importantly, the effect of co-existence of both modifications in the same nucleosome on Sir3 binding. As both H4K16ac and H3K79me are markers for euchromatin, and are deposited globally in *S. cerevisiae* (*Kimura et al., 2002*; *Ng et al., 2002*; *Suka et al., 2002*), it is highly likely that nucleosomes within euchromatic regions harbor both histone modifications at the same time. We used the piccolo histone acetyltransferase (HAT) complex to acetylate H4K16 and the methyl-lysine analog (MLA) method to generate $K_C79me3$ H3 histones (*Simon et al., 2007*), which were then reconstituted into MonoN and DiN (*Figure 5—figure supplement 1*). We chose H3K79me3 for our binding studies because it has been shown that the trimethylated state of H3K79 is the predominant in vivo state (*Frederiks et al., 2008*; *Ng et al., 2002*).

Consistent with previous results (*Johnson et al., 2009*; *Martino et al., 2009*), H4K16ac and H3K_C79me3 each decreased the affinity of Sir3 binding for MonoN by 4–5 fold, with $K_D$ values of about 4.5 μM and 5.0 μM, respectively (*Figure 5A–D*, *Table 1*). Each modification also reduced the affinity of Sir3 for DiN with $K_D$ values of 0.7 μM and 0.8 μM, respectively (*Figure 5E–F* and *Figure 5—figure supplement 2A,B*, *Table 1*), which represented about a 5-fold decrease in affinity relative to unmodified DiNs (*Figure 1E,H*). However, the combination of the two modifications inhibited Sir3 binding to MonoN and DiN, so that we could not obtain specifically shifted bands at the highest Sir3 concentration (11 μM) used in the assay (*Figure 5—figure supplement 2C,D*, *Table 1*). This observation suggests that the two modifications act together to strongly inhibit Sir3 binding. We noted slightly up-shifted bands in ac/me-modified DiN at Sir3 concentrations above 3.6 μM. As the up-shift continues to increase with higher Sir3 concentrations, but never reaches the specifically shifted band observed for unmodified Sir3-DiN complex, we surmise that the observed binding is likely nonspecific.

## Sir3wH and Sir4CC are both required for SIR complex spreading in vivo but play partially redundant roles when Sir3 is overexpressed

We next investigated the relative contribution of each the Sir3wH and the Sir4CC domains to Sir3 association with chromatin in vivo. To this end, we performed chromatin immunoprecipitation followed by high throughput sequencing (ChIP-seq) using an antibody that recognizes Sir3. The results confirmed previous studies, which have shown that each domain is required for Sir3 association with silent loci. As shown in *Figure 6A* (tracks 1–5) for the left telomere of chromosome 1 (Chr1L), the deletion of either the wH domain (*sir3ΔwH*), or a mutation of the Sir4CC that abolishes its interaction with Sir3 (*sir4-I1311N*) (*Chang et al., 2003*; *Rudner et al., 2005*), resulted in a strong loss of the Sir3 ChIP signal. Furthermore, the association of Sir3 with Chr1L in the double mutant (*sir3ΔwH*, *sir4-I1311N*) was reduced to the same level observed in *sir3Δ* cells. The results of the double mutant strain hinted at a greater loss in binding than in cells carrying each of the single mutations. However, our ChIP-seq assays were not sensitive enough to reliably detect possible differences between single and double mutant strains when Sir3 was expressed at wildtype levels (see *Figure 6—figure supplement 1A* for the result of all telomeres).

Therefore, we next performed these experiments in cells with Sir3 overexpressed from a high copy 2 micron plasmid (*SIR3 2 μ*). Sir3 overexpression has been suggested to partially bypass the requirement for Sir4 in spreading of silencing and may provide a way to assess possible Sir4-independent contribution of the wH domain to spreading (*Renauld et al., 1993*; *Strahl-Bolsinger et al., 1997*). Consistent with previous studies (*Strahl-Bolsinger et al., 1997*), we observed domains of Sir3 association at telomeres that were expanded upon Sir3 overexpression (*Figure 6*, compare tracks 1

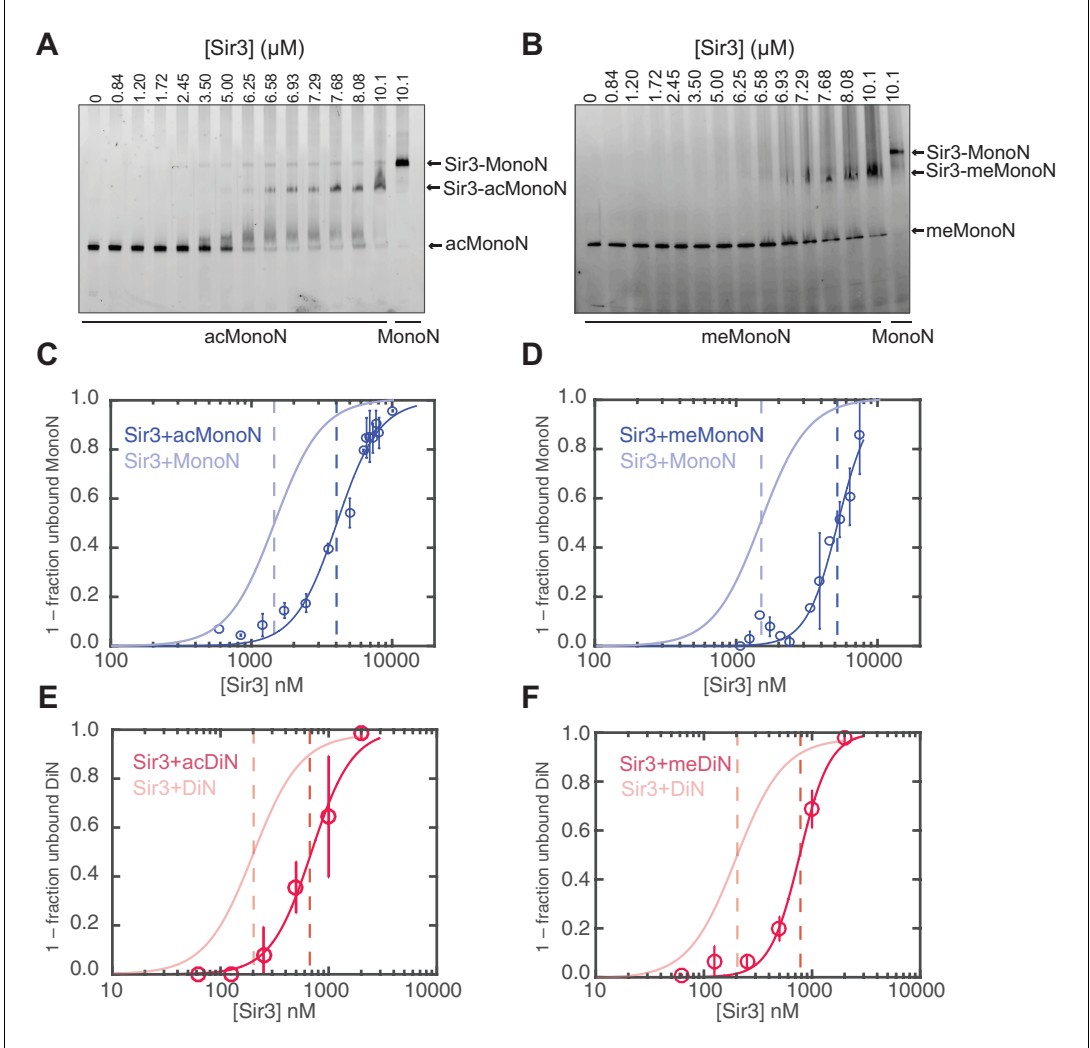

**Figure 5.** H4K16 acetylation and H3K79 methylation act together to strongly inhibit the binding of Sir3 to nucleosomes. (A, B) Comparison of Sir3 binding to unmodified and H4K16ac MonoN (A), or unmodified and H3K$_C$79me3 MonoN (B) by EMSA. We note that Sir3 bound H4K16ac and H3K$_C$79me3 MonoN and DiN shift to a lower position than Sir3 bound unmodified nucleosomes, suggesting that Sir3 binds to modified nucleosomes in a different conformation than to unmodified nucleosomes. (C) Binding curves from three experiments performed as in (A) were fitted with the Hill Equation. (D) Binding curves from three experiments performed as in (B) were fitted with the Hill Equation. (E) Comparison of Sir3 binding to unmodified and H4K16ac DiN. Binding curves from three experiments performed as in *Figure 5—figure supplement 2A* were fitted with the Hill Equation. (F) Comparison of Sir3 binding to unmodified and H3K$_C$79me3 DiN. Binding curves from three experiments performed as in *Figure 5—figure supplement 2B* were fitted with the Hill Equation. In C–F, curves in lighter colors show model fits to binding data of unmodified nucleosomes for visual comparison. Data and fits are shown in *Figure 1E*. Blue and Red dotted lines indicate the apparent K$_D$ for Sir3 binding to MonoN and DiN, respectively. See *Table 1* for parameter values. Note that the titrations of modified nucleosomes are not fully saturated at the highest concentrations tested and the calculated apparent affinities may be overestimated.

The following figure supplements are available for figure 5:

**Figure supplement 1.** Reconstitution of H4K16ac and H3K$_C$79me3 nucleosomes.

**Figure supplement 2.** Sir3 binding to singly and doubly modified H4K16 acetylated and H3K79 methylated MonoN and DiN.

and 6, *Figure 6—figure supplement 1B*). Under conditions of Sir3 overexpression, Sir3 remained detectably associated with Chr1L in both *sir3ΔwH* and *sir4-I1311N* cells, albeit at reduced levels (*Figure 6A*, tracks 7 and 8). The degree of association was higher in *sir4-I1311N* than in *sir3ΔwH* cells, suggesting that wH-mediated Sir3 dimerization played a more important role in spreading

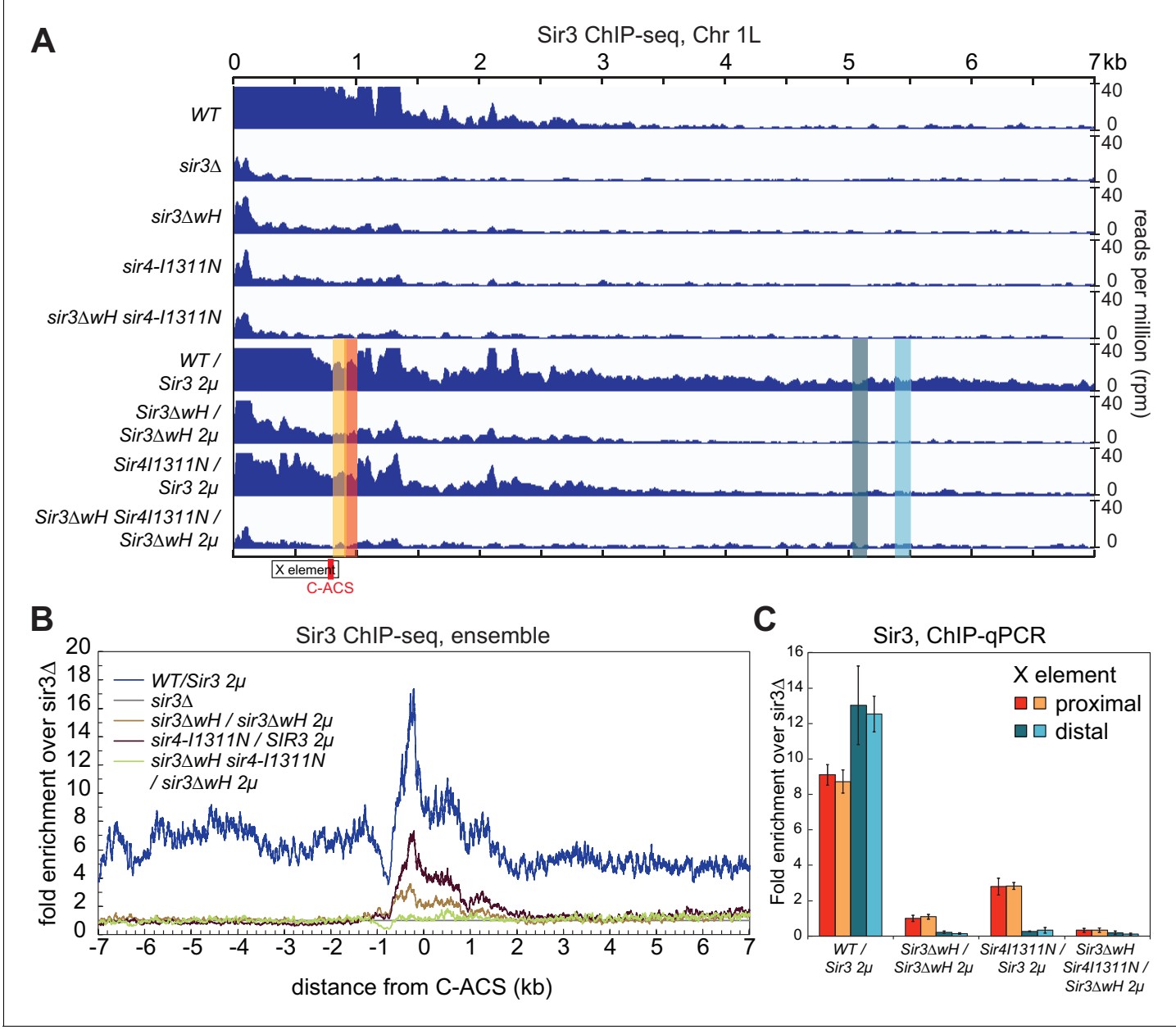

**Figure 6.** Sir3wH and Sir4CC are both required for SIR complex spreading in vivo but play partially redundant roles when Sir3 is overexpressed. (**A**) Normalized ChIP-seq read densities for Sir3 at the left telomere of chromosome 1 (Chr1L) from cells with the indicated genotypes. The core X element (black box) and the ACS within (C-ACS filled red box) are indicated below the tracks. Sir3 was overexpressed from a 2 micron plasmid (SIR3 2 μ). Colorshaded areas indicate regions assayed by ChIP-qPCR, shown in panel **C**. Full scale plots of WT and WT/Sir3 2μ strains are shown in *Figure 6— figure supplement 1B*. (**B**) Ensemble plot of Sir3 ChIP-seq signals with Sir3 overexpression, from 30 subtelomeres, excluding *TEL01R* and *TEL13R*, aligned at C-ACS. ChIP signal at all 30 subtelomeric regions was summed up, normalized to the *sir3Δ* sample, and plotted as a function of distance from C-ACS. Negative values are towards the chromosome end, and positive values are towards the centromere. Similar plot with endogenous Sir3 expression level is shown in *Figure 6—figure supplement 1A*. (**C**) Sir3 ChIP-qPCR results from cells with the indicated genotypes normalized to *sir3Δ* cells. Error bars indicate the standard deviation of three biological replicates. PCR primer sets (*Table 4*) were chosen to assay regions shown with corresponding colors in panel **A**.

The following figure supplement is available for figure 6:

**Figure supplement 1.** The requirement of Sir3wH and Sir4CC for the association of Sir3 with heterochromatin in vivo.

than the association of Sir3 with Sir4. In double mutant cells, however, Sir3 binding was reduced to background levels (*Figure 6A*, track 9). Therefore, Sir3wH and Sir4CC domains each make important contributions to Sir3 spreading, which become partially redundant when Sir3 is overexpressed.

The data for Chr1L was representative of Sir3 association with most telomeric regions. Ensemble plots of the ChIP-seq data, aligned either to chromosome ends (*Figure 6—figure supplement 1E*) or to the ACS sequence in the subtelomeric X elements (*Figure 6B* and *Figure 6—figure supplement 1F*), previously suggested to be the initiation site of SIR complex spreading in subtelomeric regions (*Ellahi et al., 2015*; *Pryde and Louis, 1999*; *Radman-Livaja et al., 2011*; *Tham and Zakian, 2002*; *Thurtle and Rine, 2014*; *Zill et al., 2010*), supported our conclusions above. Finally, Sir3 ChIP-qPCR analysis of strains shown in *Figure 6B* at four loci across chromosome 1 confirmed greater Sir3 binding deficiency in the double mutant and the more pronounced effect of Sir3ΔwH compared to Sir4I1311N mutation (*Figure 6C*). We conclude that the effects of Sir3wH and Sir4CC domains on Sir3 spreading in vivo correlate with their respective contributions to the stability of the internucleosomal Sir3 bridge in vitro.

## Discussion

Our results indicate that Sir3, the primary nucleosome-binding subunit of the SIR silencing complex, associates with chromatin via an inter-nucleosomal bridge (*Figure 7*). Sir3-Sir3 inter-nucleosomal interactions are mediated by wH dimerization domains and are further stabilized by Sir4. The inability of nucleosome binding domains in Sir3 dimers to interact with or bridge sites on the same nucleosome, or for Sir4 to bridge Sir3 molecules bound to the same nucleosome, lead us to propose that heterochromatin assembly occurs by interrupted Sir3 bridges across neighboring nucleosomes (*Figure 7*). In contrast to oligomerization or sticky end-based models, in an interrupted or discontinuous spreading mechanism Sir-Sir contacts do not extend beyond two appropriately modified nucleosomes (unacetylated H4K16, unmethylated H3K79). We propose that nucleosome pairs bearing unmodified H3K79 and H4K16 residues form the fundamental unit of Sir chromatin binding. Furthermore, the requirement for two nucleosomes in cooperative binding of the complex suggests a new rate-limiting step in nucleation of silent chromatin that may contribute to specific silencer-guided assembly of silent chromatin.

### An interrupted mechanism of heterochromatin spreading without sticky ends or oligomerization

The interrupted spreading mechanism described here is distinct from previously proposed oligomerization-based models. For example, Swi6/HP1 in *S. pombe* has been proposed to associate with chromatin via chromo shadow domain-mediated dimerization across adjacent nucleosomes as well as chromo domain-chromo domain interactions on the same nucleosome (*Canzio et al., 2011*). This 'sticky ends' mode of binding would result in the formation of continuous Swi6-Swi6 interactions across silent chromatin domains. However, oligomerization beyond stable dimers proved to be weak, even in very high (20 μM or higher) Swi6/HP1 concentrations (*Canzio et al., 2011*), and absent in mouse HP1β, even at 30 μM concentrations (*Muller-Ott et al., 2014*). Therefore, oligomerization may not contribute to the in vivo mechanism of action of HP1 proteins. In fact, subsequent re-analysis of the Swi6/HP1 binding isotherms (*Canzio et al., 2011*) revealed that a simpler model, lacking direct interactions among neighboring Swi6/HP1 dimers that were suggested in the original study, could explain the observations (*Teif et al., 2015*). Furthermore, recent studies of the mammalian HP1α and HP1β proteins suggest that they associate with chromatin primarily as nucleosome bridges (*Hiragami-Hamada et al., 2016*; *Kilic et al., 2015*).

Similarly, Sir3 can form oligomers in vitro and this oligomerization was previously suggested to mediate SIR complex spreading along chromatin (*Liou et al., 2005*; *McBryant et al., 2006*). In contrast, the discontinuous mode of binding revealed here suggests that recruitment of new SIR complexes does not rely on contacts between newly recruited and already bound complexes, but instead requires association of Sir3 with a pair of nucleosomes unmodified at H4K16 and H3K79. We note that Sir3 protein, at sub-micromolar concentrations and in buffers containing physiological salt concentrations, is found largely as a monomers and dimers (*McBryant et al., 2006*; *Swygert et al., 2014*). Therefore, the ability of Sir3 to form higher order oligomers in low salt and at high concentrations may not play a role in its cellular function. However, it remains possible that Sir3

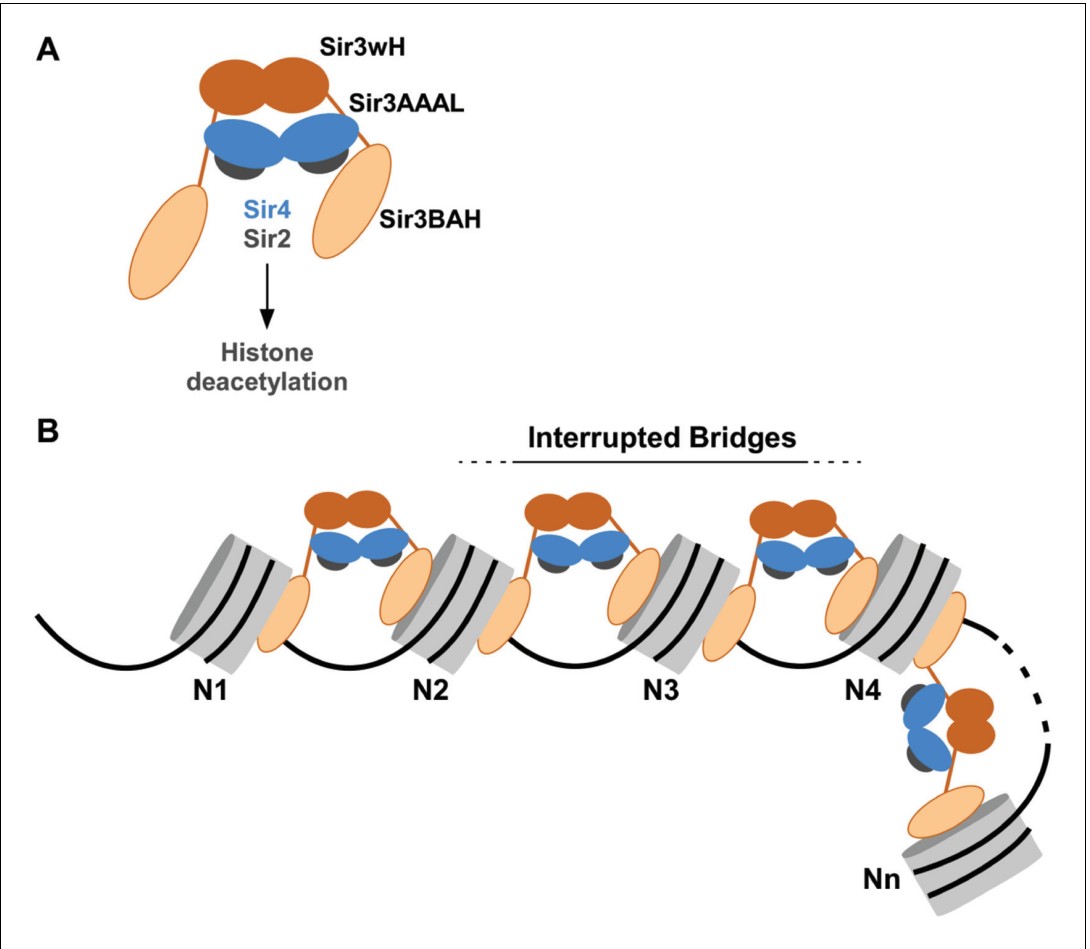

**Figure 7.** Association of the SIR complex with chromatin by interrupted Sir3 bridges across neighboring nucleosomes. (**A**) Schematic of the SIR complex indicating the location of domains in Sir3 (brown) and Sir4 (blue) dimers, and highlighting the interaction of Sir4 with Sir2. Not to scale. (**B**) Model for the association of the SIR complex with chromatin. Sir3 dimers form inter-nucleosomal bridges via cooperative association with histone H4K16 and H3K79 regions on flat nucleosome surfaces. The Sir3 bridge requires the wH domain and is further stabilized via interactions between the coiled-coil domain of Sir4 and the AAAL domain of Sir3. Spreading does not involve interaction of newly recruited Sir3/Sir4 with already bound complexes but instead requires cooperative association with a pair of H4K16 deacetylated and H3K79 unmethylated nucleosome surfaces, and thus is driven primarily by Sir2-dependent deacetylation of proximal nucleosomes. The inability of Sir3 to simultaneously interact with binding sites on the same nucleosome, requires spreading via the formation of interrupted Sir3 bridges. Sir3 may interact with immediately neighboring nucleosomes (N1 and N2, N2 and N3, N3 and N4) or more distal nucleosome pairs lacking H4K16ac/H3K79me modifications (N4 and Nn).

oligomerization is regulated by unknown factors in vivo that may be absent in our experiments, or may result from increased effective concentration of nucleosome-bound Sir3.

## Cooperative binding to appropriately modified nucleosomes can account for robust association of Sir3 with heterochromatin and its exclusion from euchromatin

Active and silent chromatin regions are associated with stereotypical patterns of histone post-translational modifications with each type of region containing several different modifications (*Ernst et al., 2011*; *Filion et al., 2010*; *Kharchenko et al., 2011*; *Taverna et al., 2007*). The combined action of multiple histone modifications potentially provides better binding specificity (*Du and Patel, 2014*; *Ruthenburg et al., 2007*). In a few cases, combinations of modification states are

recognized by the same (*Eustermann et al., 2011*; *Iwase et al., 2011*; *Moriniere et al., 2009*; *Ramón-Maiques et al., 2007*), or different domains in a single protein or different subunits of a complex (*Dhalluin et al., 1999*; *Li et al., 2006*; *2007b*; *Rothbart et al., 2013*; *Ruthenburg et al., 2011*). In budding yeast, previous studies indicate that H4K16 acetylation and H3K79 methylation each reduces Sir3 binding to nucleosomes, and therefore negatively regulate heterochromatin formation (*Braunstein et al., 1996*; *Johnson et al., 2009*; *Liou et al., 2005*; *Martino et al., 2009*; *Onishi et al., 2007*; *Swygert et al., 2014*; *van Leeuwen et al., 2002*). Moreover, recent structural analysis of Sir3BAH bound to MonoN indicates that both the H4K16 and H3K79 regions interact directly with the BAH domain (*Armache et al., 2011*; *Arnaudo et al., 2013*; *Wang et al., 2013*). Consistent with these findings, H4K16 acetylation and H3K$_C$79 trimethylation each reduces the affinity of Sir3 binding to nucleosomes around 4-fold (this study), and substitution of H4K16 with Q reduces the affinity of Sir3 for oligo-nucleosomes to a similar degree (*Swygert et al., 2014*). More strikingly, when both modifications were present in the same nucleosome, Sir3 binding affinity is reduced to a level that could not be measured in our experiments (>11 μM). Since both H3K79 methylation and H4K16 acetylation are present at high levels in euchromatic genes and absent in silent chromatin regions (*Kurdistani et al., 2004*; *van Leeuwen et al., 2002*), their combined action, together with the strong cooperative binding to surfaces on two unmodified nucleosomes, is likely to be sufficient for the specific localization of Sir3 to silencer-proximal nucleosomes lacking H3K79 and H4K16 modifications.

Cooperative modes of association, relying on multiple weak interactions rather than one strong interaction interface, are widespread in biology and contribute to modularity of regulatory networks, their robustness against noise, and their ability to display bistability (*Ptashne, 2009*; *Williamson, 2008*). The cooperative mechanism of Sir3 binding to unmodified DiN units strongly biases Sir3 binding away from association with randomly occurring deacetylated nucleosomes that may arise throughout the genome. Independent measurements have reported a wide range of Sir3 and Sir4 molecules per cell (*Chong et al., 2015*; *Gerber et al., 2003*; *Ghaemmaghami et al., 2003*; *Kulak et al., 2014*). Since Sir3 is primarily concentrated inside the nucleus (volume ~2.8 μm$^3$) (*Jorgensen et al., 2007*), Sir3 intranuclear concentration may be roughly estimated at 0.4–0.8 μM. In this range of concentrations, unmodified nucleosome pairs are ~50–80 fold more likely to be bound by Sir3 than isolated single nucleosomes (see Materials and methods). This selectivity is in large part due to binding cooperativity: even if Sir3 bound to di-nucleosomes with 10-fold higher affinity than to mono-nucleosomes, in the absence of cooperativity, di-nucleosomes would be favored over isolated nucleosomes by only ~12 fold. Therefore, cooperative association of reader proteins with nucleosome pairs may endow heterochromatin domains with robustness against random noise.

## Materials and methods

### Strains and plasmids construction

*S. cerevisae* strains and plasmids used in this study are listed in *Table 3*. All deletions and mutations were confirmed by PCR and sequencing. Epitope-tagged strains were constructed by a PCR-based gene targeting method (*Longtine et al., 1998*; *Rudner et al., 2005*).

### Protein cloning and purification

Sir3-3XFLAG and BAH-3XFLAG were purified from *S. cerevisiae* as described previously (*Buchberger et al., 2008*; *Huang et al., 2006*; *Liou et al., 2005*). Sir3ΔwH-3XFLAG was constructed by deleting the winged helix (wH) region on pDM1009 (GAL-Sir3-3XFLAG 2μ plasmid), and purified by the same FLAG purification protocol. Sir3-CBP and Sir3ΔwH-CBP were prepared by TEV cleavage of affinity purified Sir3-TAP and Sir3ΔwH-TAP proteins, followed by anion-exchange and size-exclusion chromatography as described previously (*Buchberger et al., 2008*; *Huang et al., 2006*; *Liou et al., 2005*). We tested a number of purification strategies to optimize protein yield and purity, including the addition of 500 mM KCl and 250 mM guanidine hydrochloride in size exclusion chromatography buffers. These conditions strongly decrease Sir3 oligomerization without affecting its tertiary structure (*McBryant et al., 2006*). While high salt and small amounts of chaotropic agent increased purification yield, none of our purification schemes affected the final purity or binding behavior of Sir3 to nucleosome templates. Sir4CC (1198–1358) was cloned into the pET47b(+)

**Table 3.** List of yeast strains and plasmids used in this study.

| Name | Yeast strain genotype | Source |
|---|---|---|
| W303-1a (SF1) | *MATa ade2-1 can1-100 his3-11 leu2-3,112 trp1 ura3-1 GAL* | J. Rine |
| SF4 | *sir3△::TRP1* in SF1 | J. Rine |
| DMY4350 | *sir3△wH::TRP1* in SF1 | This Study |
| ADR2973 | *sir4-I1311N* in SF1 | (*Longtine et al., 1998*; *Rudner et al., 2005*) |
| DMY4351 | *sir4-I1311N sir3△wH::TRP1* in SF1 | This Study |
| DMY3315 | W303-1a *sir3△::Kan^R hmr△E::TRP1* TELVII-L::URA3 | (*Buchberger et al., 2008*) |
| Name | Plasmid genotype | Source |
| pRS315 | *CEN/ARS LEU2* | (*Sikorski and Hieter, 1989*) |
| pJR104 (pDM602) | *SIR3* under endogenous promoter in YEp24 | (*Kimmerly and Rine, 1987*) |
| pDM1798 | *sir3△wH* under endogenous promoter in YEp24 | This Study |
| pDM832 | Sir3-3XFLAG under endogenous promoter in pRS315 | (*Buchberger et al., 2008*) |
| pDM1799 | Sir3△wH-3XFLAG under endogenous promoter in pRS315 | This Study |

All deletions and mutations were confirmed by PCR and sequencing. Epitope-tagged strains were constructed by a PCR-based gene targeting method (*Longtine et al., 1998*; *Rudner et al., 2005*).

plasmid, and the protein was purified from *E.coli* by Ni$^+$-affinity purification, followed by PreScission protease cleavage and gel filtration, to remove the His tag. A minor degradation product co-purified with Sir4CC (1198–1358). The mass Spectrometry analysis identified this fragment as Sir4 (1242–1358), which covers the entire Sir4CC core domain, and should therefore have the same Sir3 binding activity as the larger Sir4 (1198–1358). Sir4CC (1198–1358) was also cloned into pGEX6P-1, and the resulting GST-Sir4CC was affinity purified from *E.coli*. *S. cerevisiae* histones were overexpressed and purified from *E. coli* as previously described (*Johnson et al., 2009*). H3K$_C$79me3 histone was prepared as previously described (*Simon et al., 2007*). Histone H2A K120C was prepared by standard PCR-based mutagenesis and reacted with EZ-Link Maleimide-PEG2-Biotin (Thermo Fisher Scientific) following the manufacturer's protocol. The catalytic Piccolo subcomplex of the NuA4 histone acetyltransferase (HAT) complex was purified from *E. coli* as previously described (*Barrios et al., 2007*; *Johnson et al., 2009*; *Selleck et al., 2005*).

## Mono-nucleosome and nucleosome array reconstitution

MonoN and nucleosome arrays were reconstituted using gradient salt dialysis as described previously (*Luger et al., 1999*) with modifications for arrays encompassing more than 2 nucleosomes, according to Huynh et al. (*Huynh et al., 2005*). The MonoN DNA template contains the 147 bp 601 positioning sequence (*Lowary and Widom, 1998*). The array DNA templates contain defined number of direct repeats of the 601 sequence, separated by a 20 bp linkers. The 601 tetramer template also contains 20 bp DNA before and after the array. The biotinylated nucleosomal DNA template contains the 601 sequence, with an extra 20 bp linker added upstream by PCR reactions using a 5′ biotinylated primer (Integrated DNA Technologies). The Alexa-647 labeled MonoN DNA template was also made by PCR reactions using 5′ Alexa-647 labeled primer (Integrated DNA Technologies). Internucleosomal linker DNA in the *S. cerevisiae* silent chromatin regions has a heterogeneous length distribution (*Brogaard et al., 2012*; *Ravindra et al., 1999*; *Weiss and Simpson, 1998*). We chose the linker DNA to be 20 bp, which reflects the average linker DNA length in *S. cerevisiae* (*Arya et al., 2010*).

Nucleosome acetylation was carried out as described previously (*Johnson et al., 2009*). Briefly, nucleosomes were incubated with 1/10th molar ratio of the Piccolo HAT complex and 100X molar excess of acetyl-CoA in the HAT buffer (20 mM Tris.HCl, pH 8.0, 50 mM KCl, 5% glycerol, 5 mM DTT, 1 mM PMSF, and 0.5 mg/ml BSA) at 30°C for 1 hr. The completion of acetylation was assessed by the complete shift of the nucleosome band, and by quantitative Western blot using antibody against H4K16ac, where saturated signal was achieved.

### Restriction enzyme protection assay

DiN were incubated with 10U of either ScaI or AluI restriction enzyme (New England Biolabs) in 1XNEB CutSmart Buffer, at 37°C for 1 hr. The resulting digestion products were separated on native polyacrylamide gels, and visualized by staining with ethidium bromide.

### Electrophoretic mobility shift assay (EMSA)

Different amounts of Sir3 protein were incubated with 3 nM MonoN or DiN in binding buffer (25 mM Tris.HCl (pH 7.5), 50 mM NaCl, and 5 mM DTT) at 4°C for 1 hr. Samples were then run on native polyacrylamide gels, stained with SYBR Gold (Invitrogen), visualized on a Typhoon FLA7000 imager (GE Healthcare), and quantified using ImageQuant software. Sir3 binding to nucleosomes was quantified by the decrease in the intensity of the unbound nucleosome band. The apparent $K_D$ (protein concentration at transition midpoint) and Hill coefficient for each binding reaction was calculated by fitting the binding curves with the Hill Equation (see Analysis of binding cooperativity section below) using MATLAB (Mathematica).

### Biolayer interferometry (BLI) assay

BLI sensors were pre-incubated in loading buffer (20 mM Tris.HCl pH 7.5, 1 mM EDTA, 200 mM NaCl, 1 mM DTT, 0.5 mg/ml BSA, 0.02% Tween-20) before incubation with 10 nM biotinylated MonoN or DiN in the same buffer for 5–10 min. To eliminate artifacts due to surface crowding and ligand walking, nucleosome binding to sensors were first optimized to determine conditions where sensors were loaded at <25% capacity and binding behavior was insensitive to nucleosome density on sensor. All sensors for each titration experiment were loaded with the same number of nucleosomes (± 5%, as monitored by BLI loading signal). Binding experiments were performed using Octet RED384 system (Pall Life Sciences) at 30°C in the same buffer used for loading nucleosomes, except that the NaCl concentration was reduced to 50 mM. To determine the effect of nonspecific protein-sensor interactions, we measured binding signal of empty sensors in various concentrations of Sir3 as well as nucleosome-loaded sensors in buffer solution without Sir3. Nonspecific Sir3-sensor interactions gave rise to linearly rising baselines, which were subsequently subtracted from the signal using a linear extrapolation procedure. Measurements were repeated with Sir3 protein from at least two independent purifications (using FLAG or TAP tags) and two separate reconstitutions of nucleosomes templates. For larger nucleosome arrays, two biotinylation densities (all or 33% of histone octamers biotinylated) were tested in the reconstitution of nucleosome arrays to ensure that sensor immobilization does not interfere with Sir3 binding. Addition of 0.5 μM competitor DNA (salmon sperm genomic DNA physically sheared to average 150–200 bp length) did not affect Sir3 binding to nucleosomes. Therefore, average binding profiles shown in *Figure 1G–H* include experiments performed with and without competitor DNA. Furthermore, the presence of 20 bp extra linker DNA on MonoN did not affect binding of Sir3 (*Figure 1—figure supplement 3C*). Consistent with the above observations, loading of at least 10-fold higher free DNA on biosensors was necessary to obtain measurable signal changes as a result of Sir3 binding to free DNA. We therefore concluded that, the weak affinity of Sir3 for nonspecific binding to free DNA does not contribute to our nucleosome binding assays.

Association and dissociation rates and amplitudes were calculated by nonlinear least-square fitting of data with mono- or bi-exponential saturation models (see Measurement of Sir3 binding kinetics with BLI section below). The amplitudes were normalized and plotted versus protein concentration to reconstruct titration curves which were fit with the Hill equation, or when possible, with a model describing binding to identical independent sites (see Analysis of binding cooperativity section below). All model fitting procedures were performed by the nonlinear least squares method implemented in MATLAB (Mathematica).

### Analysis of binding cooperativity

The following equations were used to quantify binding experiments, as indicated in the main text.

## Cooperative binding model (Hill equation)

The cooperative binding of protein to nucleosome templates was assessed by the disappearance of free nucleosomes in EMSA or binding signal normalized to the signal at the highest protein concentration in BLI, and quantified by the Hill equation

$$f_{\text{unbound}} = \left( \frac{1}{1 + \left( \frac{[S]}{K_D} \right)^n} \right)$$

$$A_{\text{norm}} = \left( \frac{\left( \frac{[S]}{K_D} \right)^n}{1 + \left( \frac{[S]}{K_D} \right)^n} \right)$$

where $f_{\text{unbound}}$ is the fraction of free nucleosomes in EMSA, $A_{\text{norm}}$ is normalized BLI signal, [S] is free protein concentration (approximated by total protein concentration), $K_D$ is protein concentration at half saturation of the transition, and $n$ is the Hill coefficient. The Hill coefficient corresponds to the steepness of the transition with respect to the protein concentration at the midpoint of the transition and is only a phenomenological descriptor of transition cooperativity, rather than an estimate for the number of protein molecules bound to the nucleosome in the transition. $n$ values close to 1 indicate non-cooperative binding, while values larger than 1 indicate positive (favorable) cooperativity in binding of proteins to the nucleosome templates.

## Fractional saturation of two identical and independent (i.e. non-cooperative) binding sites

Binding of Sir3 to MonoN was quantified by a physical model that describes the saturation of two identical and non-interacting binding sites (two surfaces of the nucleosome core particle):

$$\Theta = \frac{1}{2} \cdot \frac{2\left( \frac{[S]}{K_D} \right) + \left( \frac{[S]}{K_D} \right)^2}{1 + \frac{[S]}{K_D} + \left( \frac{[S]}{K_D} \right)^2}$$

where $\Theta$ is the fractional saturation of binding sites on nucleosomes, [S] is free protein concentration (approximated by total protein concentration), $K_D$ is the macroscopic dissociation constant for saturation of one binding site.

## Measurement of Sir3 binding kinetics with BLI

Simple association and dissociation of Sir3 with nucleosomes is described by mono-phasic exponential functions:

Association: $Signal = A_{\text{on}} \cdot \left( 1 - e^{-k_{\text{obs}}t} \right)$

Dissociation: $Signal = A_{\text{off}} \cdot e^{-k_{\text{off}}t} + B$,

where $A_{\text{on}}$ and $k_{\text{obs}}$ are the amplitude and rate of saturation and $A_{\text{off}}$ and $k_{\text{off}}$ are the amplitude and rate of dissociation. $B$ represents the baseline signal. Protein-nucleosome association rates ($k_{\text{on}}$) were calculated from $k_{\text{obs}}$ and $k_{\text{off}}$ values

$k_{\text{on}} = \frac{k_{\text{obs}} - k_{\text{off}}}{i.[S]}$,

where $i$ is the presumed number of binding sites for the protein on nucleosome, and [S] is protein concentration.

While this model was sufficient to quantify binding of Sir3 to di-nucleosomes and larger templates (*Figure 1—figure supplement 4B,C*), we observed that it fails to capture Sir3 association and dissociation with mono-nucleosomes. Instead, a biphasic binding model, indicating two binding processes with different rates, was minimally required to fit the data (*Figure 1—figure supplement 4D,E*):

$$\begin{aligned} Signal &= A_{\text{on},1} \cdot \left( 1 - e^{-k_{\text{obs},1}t} \right) + A_{\text{on},2} \cdot \left( 1 - e^{-k_{\text{obs},2}t} \right) \\ Signal &= A_{\text{off},1} \cdot e^{-k_{\text{off},1}t} + A_{\text{off},2} \cdot e^{-k_{\text{off},2}t} + B \end{aligned}$$

Shortening of Sir3 incubation time with MonoN from 60 s to 20 s or biotinylation of nucleosomes

on DNA instead of histone H2A did not affect the biphasic dissociation of Sir3, ruling out heterogeneities in Sir3 or nucleosome preparations as the cause of biphasic binding behavior. Since Sir3 can engage multiple sites on nucleosomes through its BAH or AAAL domain (*Hecht et al., 1995*; *Liou et al., 2005*; *Martino et al., 2009*; *Onishi et al., 2007*), the two binding phases may represent different modes of Sir3-nucleosome interactions. Consistent with this hypothesis, the Sir3BAH domain alone bound to and dissociated from mono-nucleosomes with a single rate that was comparable to the fast rate of Sir3 binding and dissociation (*Figure 1—figure supplement 4F,G*, *Table 2*). More importantly, the apparent affinity of Sir3BAH for MonoN (apparent $K_D$ = 1.4 ± 0.1 µM, *Figure 1G*) closely resembled that of the fast forming fraction of the Sir3-MonoN complex ($K_D$ = 1.4 ± 0.1 µM, *Figure 1H*), while the slow-forming fraction showed a considerably lower affinity (apparent $K_D$ > 4 µM). Deletion of both AAAL and wH domains, but not the wH domain alone, caused a strong (>3 fold) decrease in Sir3-MonoN complex half-life (*Figure 1—figure supplement 4A*), confirming either a direct or synergistic contribution of AAAL domain to the interaction of Sir3 with mono-nucleosomes.

Therefore, thermodynamic, kinetic, and domain deletion experiments reveal that in addition to the BAH-mediated binding, Sir3 can engage mono-nucleosomes in other modes, most likely through the AAAL domain (*Ehrentraut et al., 2011*; *Hecht et al., 1995*), which are precluded in binding to larger nucleosome arrays ([Sir3] <0.5 µM). Therefore, we compared binding of Sir3 to larger nucleosome templates with its fast phase of binding to MonoN (*Figure 1H*).

## Non-covalent nucleosome crosslinking assays

Nucleosomes assembled with biotinylated DNA were conjugated to Dynabeads M-280 streptavidin (Invitrogen) at RT for 1 hr with rotation, using 36 µl of beads slurry per µg of nucleosomes in the binding buffer (20 mM Tris.HCl (pH 7.5), 0.3 mM EDTA, 50 mM NaCl, 10% glycerol, 5 mM DTT, 1 mg/ml BSA, and 0.02% NP–40). Bead-conjugated nucleosomes were washed, and resuspended in equal volume of binding buffer as the initial volume of beads taken. Equal amount of conjugated nucleosomes, in a final concentration of 100 nM, was added to tubes containing Sir3, Sir3/Sir4CC, Sir3ΔwH, Sir3ΔwH/Sir4CC, Sir4CC, or buffer alone, and incubated with rotation at 4°C for 1 hr. The concentration of Sir3 proteins was 2 µM in the case of high protein concentration binding assay, and 200 nM in the case of low protein concentration binding assay. Sir4CC was in 2X molar excess of Sir3 proteins. Subsequently, Alexa-647 labeled nucleosomes were added into each reaction at a final concentration of 100 nM, and reactions were incubated for another 1 hr at 4°C. Finally, the beads were washed twice in the binding buffer before magnetic concentration. Alexa-647 nucleosomal DNA from the crosslinked nucleosomes was stripped from the beads with 2M NaCl, separated on native polyacrylamide gels, and quantified by the fluorescent intensity of the band.

## ChIP-seq

Cells were cultured overnight in YEPD medium, or selective media for cells harboring overexpression plasmids (YEp24 2 µ plasmid with Sir3 or Sir3ΔwH expressed from Sir3 endogenous promoter), diluted into fresh media to $OD_{600}$ = 0.4, and harvested at late log phase ($OD_{600}$ = 1.5). Cells were fixed with 1% formaldehyde for 15 min at room temperature (RT), then quenched with 130 mM glycine for 5 min at RT, harvested by centrifugation, washed twice with TBS (50 mM Tris.HCl pH 7.6, 150 mM NaCl), and flash frozen. Cell pellets were resuspended in 600 µl lysis buffer (50 mM HEPES-KOH pH 7.5, 150 mM NaCl, 1 mM EDTA, 1% Triton X-100, 0.1% Na-Deoxycholate, 0.1% SDS, 1 mM PMSF, protease inhibitor tablet (Roche)), and disrupted by bead beating (MagNA Lyser, Roche) for 6 × 30 s at 4500 rpm with 0.5 mm glass beads. Tubes were punctured and the flow-through was collected in a new tube by centrifugation. After sonication for 3 × 20 s at 40% amplitude (Branson Digital Sonifier), the extract was centrifuged (Eppendorf 5415R) for 15 min at 13,000 rpm. The soluble chromatin was then transferred to a fresh tube. Sir3 antibody (*Rudner et al., 2005*) was preincubated with washed Dynabeads Protein A, and for each immunoprecipitation, 2 µg antibody coupled to 100 µl beads was added to soluble chromatin. Samples were incubated for 2 hr at 4°C with rotation, after which the beads were collected on magnetic stands, and washed 3 times with 1 ml lysis buffer and once with 1 ml TE, and eluted with 100 µl preheated buffer (50 mM Tris.HCl pH 8.0, 10 mM EDTA, 1% SDS) at 65°C for 15 min. The eluate was collected, and 150 µl 1XTE/0.67% SDS was added. Immunoprecipitated samples were incubated overnight at 65°C to reverse crosslink, and

treated with 50 μg RNase A at 37°C for 1 hr. 5 μl proteinase K (Roche) was added and incubation was continued at 55°C for 1 hr. Samples were purified using a PCR purification kit (Qiagen).

Libraries for Illumina sequencing were constructed as described previously (*Wong et al., 2013*), starting with ~5 ng of immunoprecipitated DNA fragments. Each library was generated with custom-made adapters carrying unique barcode sequences at the ligating end. Barcoded libraries were mixed and sequenced with Illumina HiSeq 2500. Raw reads were separated according to their barcodes and mapped to the *S. cerevisiae* S288C genome using Bowtie. Mapped reads were normalized to reads per million and visualized in IGV.

Ensemble plots aligned at chromosome ends were generated by aligning all 30 telomeres, excluding *TEL01R* and *TEL13R,* at chromosome ends, and calculating the total ChIP-seq signal across 14 kb regions towards the centromere. The cumulative ChIP-seq reads was then normalized, on a per-base basis, to that of the Δ*sir3* sample. Ensemble plots aligned at the ACS within the core X element (C-ACS) were generated in a similar manner, except that each telomere was aligned at C-ACS, and total ChIP-seq signal was computed for 14 kb in each direction.

## Quantitative PCR

ChIP was performed essentially as described in the ChIP-seq above, except for the modification described below. Extracts were sonicated for 3 × 20 s at 50% amplitude using a sonicator (Branson Digital Sonifier). After centrifugation for 15 min at 13,000 rpm, the soluble chromatin was transferred to a fresh tube and normalized for protein concentration by the Bradford assay. For each immunoprecipitation, 2 μg Sir3 antibody coupled to 30 μl Dynabeads Protein A was used. Immunoprecipitation, washes, elution and reverse crosslinking were performed as described in the ChIP-seq section. 60 μg glycogen, 44 μl of 5M LiCl and 250 μl TE were added and the samples were extracted with phenol/chloroform, and ethanol precipitated. DNA was resuspended in 100 μl of 10 mM Tris pH 7.5, 50 mM NaCl. 2.5 μl of immunoprecipitated DNA was used for qPCR. Primers used are listed in *Table 4*. qPCR was performed in the presence of SYBR Green using an Applied Biosystems 7900HT light cycler. Fold enrichments were calculated using the ΔCT method and average values of three biological replicates were normalized to the *cup1+* gene. Enrichment relative to the control (*sir3Δ*) was calculated after normalization.

## Equilibrium distribution of Sir3 among H4K16 acetylated and unmodified nucleosomes

The Boltzmann distribution describes the probability distribution of a system consisting of multiple free energy states. For a system consisting of *n* states, the probability $p_i$ of a given state *i* is calculated as

$$p_i = \frac{e^{\frac{-\Delta G_i}{kT}}}{\sum_{i=1}^{n} e^{\frac{-\Delta G_i}{kT}}},$$

where $\Delta G_i$ is the Gibbs free energy of the state *i* with respect to a common reference state, k is the Boltzmann constant, and T is absolute temperature.

In our analysis, the system consists of three types of Sir3-nucleosome complexes: unmodified di-nucleosome units, unmodified isolated MonoN, and acetylated nucleosomes. For each complex state (*i*), the free energy (relative to the unbound state) at a given Sir3 concentration [*S*] is

$$\Delta G_i = RTlnK_D = RTln\left(\frac{f_{\text{unbound}}}{1 - f_{\text{unbound}}}\right) = -n_i RTln\left(\frac{[S]}{K_{D,i}}\right),$$

where $K_{D,i}$ and $n_i$ are experimentally determined apparent dissociation constants and Hill coefficients, respectively (*Table 1*).

## Acknowledgements

We thank Rebecca Mathew and other members of the Moazed laboratory, Sarah Woodson, Ian Dodd, and Kim Sneppen for discussion, Feng Wang for help in protein purification, Aaron Johnson for construction of the H3K79C plasmid, Ruby Yu for help with ChIP-Seq data analysis and Python

**Table 4.** List of qChIP PCR primers used in this study.

| Primer | Sequence |
| --- | --- |
| RB139 (C-ACS distal 1, F) | TTC TGC CCA TAC GAT ACC T |
| RB140 (C-ACS distal 1, R) | AGT TAC GCG TGC TAC ATT AC |
| RB141 (C-ACS distal 2, F) | GTT CTA CTG ACA GGA TGG AAT AG |
| RB142 (C-ACS distal 2, R) | GTG AAG GAG GGC ATG AAA T |
| RB143 (C-ACS proximal 1, F) | CGT ACT TAC ACA GGC CAT AC |
| RB144 (C-ACS proximal 1, R) | GTT TGA GCC ACT ACC GTA TTA |
| RB145 (C-ACS proximal 2, F) | CTT GTG GTA GCA ACA CTA TCA |
| RB146 (C-ACS proximal 2, R) | GGC CTG TGT AAG TAC GAA AT |

scripts, Kelly Arnett and the Center for Macromolecular Interactions at Harvard Medical School for instrument support. This work was supported by NIH grant GM61641 (DM). DM is an investigator of the Howard Hughes Medical Institute.

# Additional information

## Funding

| Funder | Grant reference number | Author |
| --- | --- | --- |
| National Institutes of Health | GM61641 | Danesh Moazed |
| Howard Hughes Medical Institute | | Danesh Moazed |

The funders had no role in study design, data collection and interpretation, or the decision to submit the work for publication.

## Author contributions

RB, CL, Conception and design, Acquisition of data, Analysis and interpretation of data, Drafting or revising the article; MAC, GJ, Acquisition of data, Analysis and interpretation of data; NI, Acquisition of data, Analysis and interpretation of data, Drafting or revising the article; DM, Conception and design, Analysis and interpretation of data, Drafting or revising the article

## Author ORCIDs

Reza Behrouzi, http://orcid.org/0000-0003-3064-9743
Danesh Moazed, http://orcid.org/0000-0003-0321-6221

# Additional files

## Major datasets

The following dataset was generated:

| Author(s) | Year | Dataset title | Dataset URL | Database, license, and accessibility information |
| --- | --- | --- | --- | --- |
| Lu C, Jih GT, Moazed D | 2017 | Cooperative Associations with Sites on Different Nucleosomes Mediate Heterochromatin Spreading | https://www.ncbi.nlm.nih.gov/geo/query/acc.cgi?acc=GSE76553 | Publicly available at the NCBI Gene Expression Omnibus (Accession no: GSE76553) |

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
