## [Decision Letter]

Thank you for submitting your article "Heterochromatin assembly by interrupted Sir3 bridges across neighboring nucleosomes" for consideration by *eLife*. Your article has been favorably evaluated by Jessica Tyler (Senior Editor) and three reviewers, one of whom, Asifa Akhtar (Reviewer #1), is a member of our Board of Reviewing Editors.

The reviewers have discussed the reviews with one another and the Reviewing Editor has drafted this decision to help you prepare a revised submission.

The three reviewers appreciated that the study provides a thorough biochemical and biophysical characterization of the Sir proteins to defined nucleosomal substrates. In particular, the work provides novel insight into the preferred chromatin binding substrate of yeast SIR proteins which is shown to be dinucleosomes that are bound in a 'bridging' mode and that high affinity binding requires that these nucleosomes lack both H4K16ac and H3K79me3.

Overall, the reviewers consider that this manuscript would be appropriate for publication after a round of revision. Reviewers have made a number of suggestions that we would like you to address before the manuscript can be further considered.

The authors claim, that the effects they nicely characterize in vitro are linked to a discontinuous spreading mode in vivo. The discontinuous model implies, that spreading does not rely on oligomerization, which has been proposed for many chromatin effectors. However, the only in vivo evidence presented is the ChIP-seq experiments, which is not a direct proof for the discontinuous model. Would it be possible to use another experiment as an alternative to analyze the dynamics/cooperativity of binding in vivo? Is Sir3 as dynamic as it was suggested for HP1? (Cheutin et al., Stunnenberg et al., Kilic et al.).

The authors show that dimerization of Sir3 via the winged helix (which was demonstrated earlier by the Gasser group) is necessary for this bridging. However, the effect of enhancement by the Sir4 CC is not very pronounced (Figure 3). It appears that the authors ignore throughout the paper that there is a second nucleosome-binding module in Sir3 besides the BAH-domain, namely the AAA domain. How does this affect models of Sir3 binding to mono-nuc or di-nucs? Please explain and discuss the appropriate literature.

Other comments:

Given the experiments performed in the manuscript with the different Sir domains (e.g. BAH domain), it would be important to explain the relevance of this domain for nucleosome binding in the Introduction / text. Also it would be helpful for the non-expert reader to provide a scheme with the Sir3 domains.

In some of the Figures concentrations are shown in μm and nM respectively. It would be easier to understand if concentrations are labeled consistently (best in nM).

"ScaI digestion of the linker DNA in the DiN resulted in a reduction inbinding affinity to that observed for the NCP (Figure 1—figure supplement 1), indicating that specific binding to DiN, not extra DNA content, is responsible for higher affinity binding." This claim can be further substantiated by performing a competition experiment with naked DNA.

Figure 1—figure supplement 3 – Please show the raw data (kinetics) of Sir3 binding to H4K16ac nucleosomes (BLI-data)

The differences presented in Figure 6 seem very minimal / small. As these minimal differences are important for the biological relevance of claims made in the paper, it is required to substantiate them with ChIP-qPCR experiments. Is the ChIP-seq profile an average of several replicates?

How does the fact that Sir3 oligomerizes in vitro affect the interpretation of the binding data?

In Figure 5, were the curves for Sir3 binding to unmodified nucs drawn from a single data point?

The whole paper uses Sir3 purified from yeast for the binding assays. How can we be sure that no other co-purifying factors are affecting the assay (despite the gel shown in the supplement, which shows some low molecular weight bands)? How about modifications on Sir3?

In the last sentence, it would be helpful if the authors could spell out what they mean with "properly modified nucleosomes" and "appropriately modified nucleosomes". i.e. the authors should spell out that "properly modified nucleosomes" means nucleosomes lacking H4K16ac and H3K79me3.

---

## [Author Response]

[…]

*Overall, the reviewers consider that this manuscript would be appropriate for publication after a round of revision. Reviewers have made a number of suggestions that we would like you to address before the manuscript can be further considered.*

We thank the reviewers and editors for their thoughtful comments and constructive criticism. Below, we provide a point-by-point response and indicate corresponding modifications in the manuscript.

In addition to the revisions prompted by the reviewers’ comments, we have also modified

1) Figure 1—figure supplement 1 to improve the appearance of the tetra-nucleosome gel.

2) Figure 2 to include additional Sir3+TetraN replicates and also remove some visual artifacts, where some error bars appeared as lines.

*The authors claim, that the effects they nicely characterize* in vitro *are linked to a discontinuous spreading mode* in vivo*. The discontinuous model implies, that spreading does not rely on oligomerization, which has been proposed for many chromatin effectors. However, the only* in vivo *evidence presented is the ChIP-seq experiments, which is not a direct proof for the discontinuous model. Would it be possible to use another experiment as an alternative to analyze the dynamics/cooperativity of binding* in vivo*? Is Sir3 as dynamic as it was suggested for HP1? (Cheutin et al., Stunnenberg et al., Kilic et al.).*

We agree with the reviewer that additional tests are required to test whether the proposed mechanism of spreading in vitro reflects in vivo heterochromatin spreading. We modified the text to indicate this point (subsection “An interrupted mechanism of heterochromatin spreading without sticky ends or oligomerization” Discussion). However, at the present time, we are unaware of any experimental method that can directly probe the molecular mechanism of spreading with the resolution necessary to differentiate oligomerization and interrupted spreading models in vivo. While it is interesting to measure the in vivo dynamics of Sir3 binding to chromatin as suggested by the reviewer, such measurements cannot prove or refute any of the spreading models discussed in the literature. For example, in vivo binding dynamics of HP1 measured by Cheutin et al., Stunnenberg et al., and Kilic et al. is *not* incompatible with either of the models proposed by Canzio et al. (2011) or Kilic et al. (2015). The latter work disfavors oligomerization model as a result of in vitro single molecule measurements. Hence, we believe such measurements lie outside of the scope of this work.

We note that the discontinuous assembly model, particularly in combination with in vivo Sir3 measurements, is consistent with non-uniform ChIP-seq profiles at heterochromatic mating type loci (Thurtle and Rine, 2014), whereas oligomerization (at least in the strict sense of the word) predicts a uniform Sir3 distribution across the locus. (subsection “An interrupted mechanism of heterochromatin spreading without sticky ends or oligomerization”, Discussion).

*The authors show that dimerization of Sir3 via the winged helix (which was demonstrated earlier by the Gasser group) is necessary for this bridging. However, the effect of enhancement by the Sir4 CC is not very pronounced (Figure 3). It appears that the authors ignore throughout the paper that there is a second nucleosome-binding module in Sir3 besides the BAH-domain, namely the AAA domain. How does this affect models of Sir3 binding to mono-nuc or di-nucs? Please explain and discuss the appropriate literature.*

We acknowledge that the function of AAAL domain has been insufficiently discussed in the main text and thank the reviewers for pointing it out. We now discuss this topic on several occasions in the manuscript. In summary, we cite the previous work showing AAAL interacts with chromatin, but point out our previous measurements showing that this interaction has a much lower affinity than BAH-nucleosome interaction (Introduction). In light of the data presented in this manuscript, we argue that AAAL domain has a clear (synergistic or additive) role in stabilizing the binding of Sir3 to nucleosome, but does not directly affect binding cooperativity to di-nucleosomes (Results).

*Other comments:*

*Given the experiments performed in the manuscript with the different Sir domains (e.g. BAH domain), it would be important to explain the relevance of this domain for nucleosome binding in the Introduction / text. Also it would be helpful for the non-expert reader to provide a scheme with the Sir3 domains.*

We have described the relative contribution of each Sir3 domain to histone and nucleosomes more fully in the Introduction (fourth paragraph) and have referred the reader to the schematic diagram of the domains in Figure 1.

*In some of the Figures concentrations are shown in μm and nM respectively. It would be easier to understand if concentrations are labeled consistently (best in nM).*

All binding curves are now plotted with units of nM.

*"ScaI digestion of the linker DNA in the DiN resulted in a reduction inbinding affinity to that observed for the NCP (Figure 1—figure supplement 1), indicating that specific binding to DiN, not extra DNA content, is responsible for higher affinity binding." This claim can be further substantiated by performing a competition experiment with naked DNA.*

We had performed other assays that demonstrated the insignificance of linker DNA binding in our assay conditions (Figure 1—figure supplement 3) and now more fully describe these experiments (subsection “Biolayer Interferometry (BLI) assay”, Methods). Briefly, Figure 1—figure supplement 3 shows indistinguishable binding to nucleosomes with and without 20bp extra linker. Furthermore, in our initial BLI measurements of Sir3 binding to mono and di-nucleosomes, 0.5µM competitor DNA (salmon sperm genomic DNA sheared to average 150-200 bp length) was added to minimize nonspecific binding potential to sensors, but was omitted later as it proved to be unnecessary. Average binding profiles shown in Figure 1 include experiments performed with and without competitor DNA, as they did not show any difference.

We also attempted to measure the affinity of Sir3 for binding to long DNA templates in the absence of nucleosomes. However, about 50 fold higher free DNA than what was present in nucleosome binding assays was needed to obtain measurable binding signal with BLI assays. We therefore concluded that binding to free DNA does not contribute to binding signal measured in our assays, which is fully consistent with previous observations by Swygert et al. (2015).

*Figure 1—figure supplement 3 – Please show the raw data (kinetics) of Sir3 binding to H4K16ac nucleosomes (BLI-data)*

Figure 1—figure supplement 3 is updated to show the raw binding data that was used in making the histogram.

*The differences presented in Figure 6 seem very minimal / small. As these minimal differences are important for the biological relevance of claims made in the paper, it is required to substantiate them with ChIP-qPCR experiments. Is the ChIP-seq profile an average of several replicates?*

As suggested by the reviewers, we have updated Figure 6 to show ChIP-qPCR data for strains overexpressing Sir3. We have also modified the text (Results) and the layout of Figure 6 to remove a possible point of confusion regarding the conclusions of this section.

We completely agree with the reviewers that our ChIP-seq results are not sensitive enough to detect potential differences between single and double mutant strains, when Sir3 is expressed at its normal level (Figure 6—figure supplement 1). The conclusion regarding the distinct effects of Sir3∆wH and Sir4I1311N mutations are drawn from strains overexpressing Sir3 (Figure 6). In these strains, the spreading defects caused by each single mutation are very strong as they drastically limit the range and level of Sir3-bound domains around telomeric nucleation sites. In the double mutant, heterochromatin is completely defective for spreading, which is distinguishable from the single mutants.

*How does the fact that Sir3 oligomerizes* in vitro *affect the interpretation of the binding data?*

As discussed (Discussion), there is very little or no Sir3 oligomerization in the conditions used in our assays. Our observations are consistent with recent SV-AUC measurements by Swygert et al. (2015) that used very similar assay conditions. Sir3 oligomerization in some of earlier studies appears to be due to the very low salt concentrations used in reaction buffers. We have not explored the effect of Sir3 oligomerization in mid to high μM concentrations as nucleosome binding and in vivo Sir3 concentration measurements fall well short of such concentrations.

*In Figure 5, were the curves for Sir3 binding to unmodified nucs drawn from a single data point?*

We clarified the figure legends to remove the confusion. To keep the panel visually simple, the binding curves for unmodified nucleosomes show only the model fitted to the data. The data points and the model are shown in Figure 1.

*The whole paper uses Sir3 purified from yeast for the binding assays. How can we be sure that no other co-purifying factors are affecting the assay (despite the gel shown in the supplement, which shows some low molecular weight bands)? How about modifications on Sir3?*

We thank the reviewers for highlighting this aspect. We revised Results and Methods section to review relevant literature and add further details about Sir3 purification. Briefly, we used Sir3 purified from yeast because N-terminal acetylation of Sir3 (which is absent in bacterial expression) is necessary for efficient binding of the BAH domain to nucleosomes. We now point out that different affinity purifications (through TAP or FLAG tags) combined with a variety of FPLC steps (gel filtration in 150 mM or 500 mM NaCl, with or without 250 mM guanidine hydrochloride, ion exchange chromatography) all resulted in Sir3 with identical binding profiles to mono and di-nucleosomes. These results, in addition to the strong difference in band densities on protein gels, render the effect of any additional protein factors very unlikely.

We also note that the majority of smaller protein bands on the gel correspond to Sir3 degradation products (unpublished results and McBryant et al. (2006)).

*In the last sentence, it would be helpful if the authors could spell out what they mean with "properly modified nucleosomes" and "appropriately modified nucleosomes". i.e. the authors should spell out that "properly modified nucleosomes" means nucleosomes lacking H4K16ac and H3K79me3.*

Revised text as indicated (Abstract; Introduction; Discussion; Figure 7 legend).